# rPPG-Toolbox: Deep Remote PPG Toolbox

**Xin Liu[1], Girish Narayanswamy[1*], Akshay Paruchuri[2*], Xiaoyu Zhang[3], Jiankai Tang[3],**
**Yuzhe Zhang[3], Roni Sengupta[2], Shwetak Patel[1], Yuntao Wang[3], Daniel McDuff[1]**
University of Washington Seattle [1]
University of North Carolina at Chapel Hill [2]
Tsinghua University [3]
`{xliu0, girishvn, dmcudff}@cs.washington.edu, akshay@cs.unc.edu`
∗ Equal Contribution

## Abstract

Camera-based physiological measurement is a fast growing field of computer vision. Remote photoplethysmography (rPPG) utilizes imaging devices (e.g., cameras) to measure the peripheral blood volume pulse (BVP), and enables cardiac measurement via webcams and smartphones. However, the task is non-trivial with important pre-processing, modeling, and post-processing steps required to obtain state-of-the-art results. Replication of results and benchmarking of new models is critical for scientific progress; however, as with many other applications of deep learning, reliable codebases are not easy to find or use. We present a comprehensive toolbox, rPPG-Toolbox, that contains unsupervised and supervised rPPG models with support for public benchmark datasets, data augmentation, and systematic evaluation: `https://github.com/ubicomplab/rPPG-Toolbox`

## 1 Introduction

The vision of ubiquitous computing is to embed computation into everyday objects to enable them to perform useful tasks. The sensing of physiological vital signs is one such task and plays an important role in how health is understood and managed. Cameras are both ubiquitous and versatile sensors, and the transformation of cameras into accurate health sensors has the potential to make the measurement of health signals more comfortable and accessible. Examples of the applications of this technology include systems for monitoring neonates [1], dialysis patients [2], and the detection of arrhythmias [3].

Building on advances in computer vision, camera-based measurement of physiological vitals signs has developed into a research field of its own [**?** ]. Researchers have developed methods for measuring cardiac and pulmonary signals by analyzing skin pixel changes over time. Recently, several companies have been granted FDA De Novo status for products that use software algorithms to analyze video and estimate pulse rate, heart rate, respiratory rate and/or breathing rate[12].

There are hundreds of computational architectures that have been proposed for the measurement of cardiopulmonary signals. Unsupervised signal processing methods leverage techniques such as Independent Component Analysis (ICA) or Principal Component Analysis (PCA) and assumptions about the periodicity or structure of the underlying blood volume pulse waveform. Neural network architectures can be trained in a supervised fashion using videos with synchronized gold-standard ground truth signals [4, 5, 6, 7]. Innovative data generation [8] and augmentation [9], meta-learning for personalization [10, 11], federated learning [12], and unsupervised pretraining [13, 14, 15, 16] have been widely explored in the field of camera-based physiological sensing and have led

---

[1] `https://www.accessdata.fda.gov/cdrh_docs/reviews/DEN200019.pdf`
[2] `https://www.accessdata.fda.gov/cdrh_docs/reviews/DEN200038.pdf`

37th Conference on Neural Information Processing Systems (NeurIPS 2023) Track on Datasets and Benchmarks.

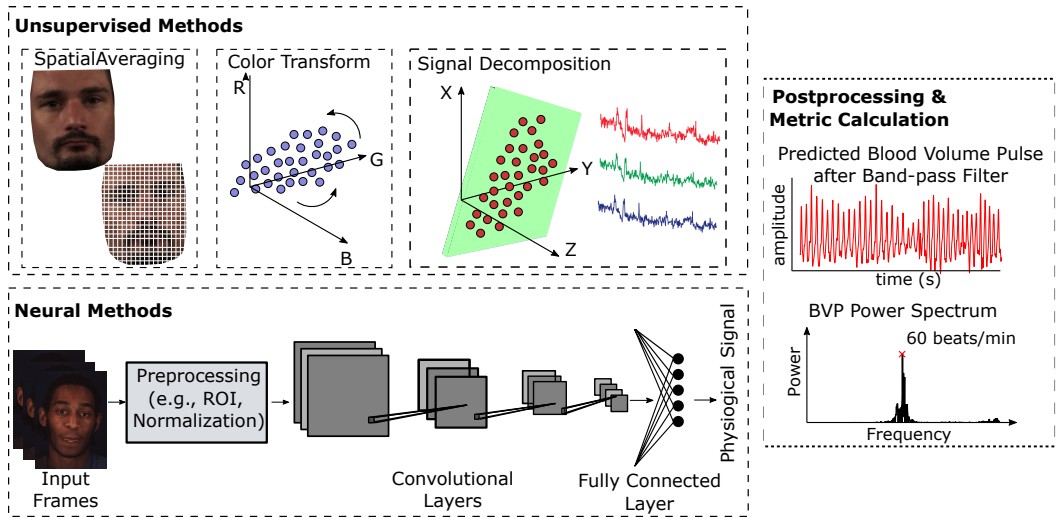

Figure 1: **rPPG Pipeline.** An example of the components of an rPPG pipeline including preprocessing, training, inference, and evaluation.

to significant improvements in state-of-the-art performance. Further information regarding the background, algorithms, and potential applications of rPPG are included in the Appendix-B and C.

However, standardization in the field is still severely lacking. Based on our review of literature in the space, we identified four issues that have hindered the interpretation of results in many papers. First, and perhaps most obviously, a number of the published works are not accompanied by public code. While publishing code repositories with papers is now fairly common in the machine learning and computer vision research communities, it is far less common in the field of camera-based physiological sensing. While there are reasons that it might be difficult to release datasets (e.g., medical data privacy), we cannot find good arguments for not releasing code. Second, many papers do not compare to previously published methods in an "apples-to-apples" fashion. This point is a little more subtle, but rather than performing systematic side-by-side comparisons between methods, the papers compare to numerical results from previous work, even if the training sets and/or test sets are not identical (e.g., test samples were filtered because they were deemed to not have reliable labels). Unfortunately, this often makes it unclear if performance differences are due to data, pre-processing steps, model design, post-processing, training schemes and hardware specifications, or a combination of the aforementioned. Continuing this thread, the third flaw is that papers use pre- and post-processing steps that are not adequately described. Finally, different researchers compute the "labels" (e.g., heart rate) using their own methods from the contact PPG or ECG time-series data. Differences in these methods lead to different labels and a fundamental issue when it comes to benchmarking performance. When combined, the aforementioned issues make it very difficult to draw conclusions from the literature about the optimal choices for the design of rPPG systems.

Open source codes allow researchers to compare novel approaches to consistent baselines without ambiguity regarding the implementation or parameters used. This transparency is important as subsequent research invariably builds on prior state-of-the-art. Implementing a prior method from a paper, even if clearly written, can be difficult. Furthermore, it is an inefficient use of time for many researcher to re-implement all baseline methods. In an effort to address this, several open source toolboxes have been released for camera-based physiological sensing. These toolboxes have been a significant contribution to the community and provide implementations of methods and models [17, 18, 19]; however, they are also incomplete. McDuff and Blackford [17][3] implemented a set of source separation methods (Green, ICA, CHROM, POS) and Pilz [19] published the PPGI-Toolbox[4] containing implementations of Green, SSR, POS, Local Group Invariance (LGI), Diffusion Process (DP) and Riemannian-PPGI (SPH) models. These toolboxes are implemented in MATLAB (e.g., [17]) and do not contain examples of supervised methods. Python and supervised neural models are now the focus of a large majority of computer vision and deep learning research. There are

---

[3]https://github.com/danmcduff/iphys-toolbox
[4]https://github.com/partofthestars/PPGI-Toolbox

Table 1: **Comparison of rPPG Toolboxes.** Comparison of rPPG-Toolbox with existing toolboxes in camera-based physiological measurement.

| Toolbox | Dataset Support | Unsup. Eval | DNN Training | DNN Eval |
|---|---|---|---|---|
| iPhys-Toolbox [20] | ✗ | ✓ | ✗ | ✗ |
| PPG-I Toolbox [19] | ✗ | ✓ | ✗ | ✗ |
| pyVHR [18, 21] | ✓ | ✓ | ✗ | ✓ |
| rPPG-Toolbox (Ours) | ✓ | ✓ | ✓ | ✓ |

Unsup. = Unsupervised learning methods, DNN = Deep neural network methods.

several implementations of popular signal processing methods in Python: Bob.rrpg.base[5] includes implementations of CHROM, SSR and Boccignone et al. [18] released code for Green, CHROM, ICA, LGI, PBV, PCA, and POS. Several published papers have included links to code; however, often this is only inference code and not training code for neural models. Without providing training code for neural networks, it is challenging for researchers to conduct end-to-end reproducible experiments and build on existing research.

In this paper, we present an end-to-end toolbox[6] for camera-based physiological measurement. This toolbox includes: 1) support for six public datasets, 2) pre-processing code to format the datasets for training neural models, 3) implementations of six neural model architectures and six unsupervised learning methods, 4) evaluation and inference pipelines for supervised and unsupervised learning methods for reproducibility and 5) enabling advanced neural training and inference such as weakly supervised pseudo labels, motion augmentation and multitask learning. We use this toolbox to publish clear and reproducible benchmarks that we hope will provide a foundation for the community to compare methods in a more rigorous and informative manner.

## 2   Related Work

In the field of remote PPG sensing, there are three significant open-source toolboxes (documented in Table 1):

**iPhys-Toolbox** [17]: An open-sourced toolbox written in MATLAB that is comprised of implementations of numerous algorithms for rPPG sensing. It empowers researchers to present results on their datasets using public, standard implementations of baseline methods, ensuring transparency of parameters. This toolbox incorporates a wide range of widely employed baseline methods; however, it falls short on Python support, public dataset data loaders, and neural network training and evaluation.

**PPG-I Toolbox** [19]: This toolbox provides MATLAB implementations, specifically for six unsupervised signal separation models. It incorporates four evaluation metrics, including Pearson correlation, RMSE/MSE, SNR, and Bland-Altman plots. However, similar to the iPhys-Toolbox, it lacks support for public dataset data loading and neural network training and evaluation.

**pyVHR [21]**: The most recent in the field, this toolbox adopts Python instead of MATLAB. While it offers ample support for numerous unsupervised methods, its capabilities are limited when it comes to modern neural networks. Notably, pyVHR supports only two neural networks for inference, and none for model training. This omission can be a roadblock for researchers aiming to reproduce and further advance state-of-the-art neural methods.

## 3   The rPPG-Toolbox

To address the gaps in the current tooling and to promote reproducibility and clearer benchmarking within the camera-based physiological measurement (rPPG) community, we present an open-source toolbox designed to support six public datasets, six unsupervised methods and six neural methods for data preprocessing, neural model training and evaluation, and further analysis.

### 3.1   Datasets

The toolbox includes pre-processing code that converts six public datasets into a form amenable for training with neural models. The standard form for the videos we select includes raw frames and

---

[5]https://pypi.org/project/bob.rppg.base/
[6]https://github.com/ubicomplab/rPPG-Toolbox

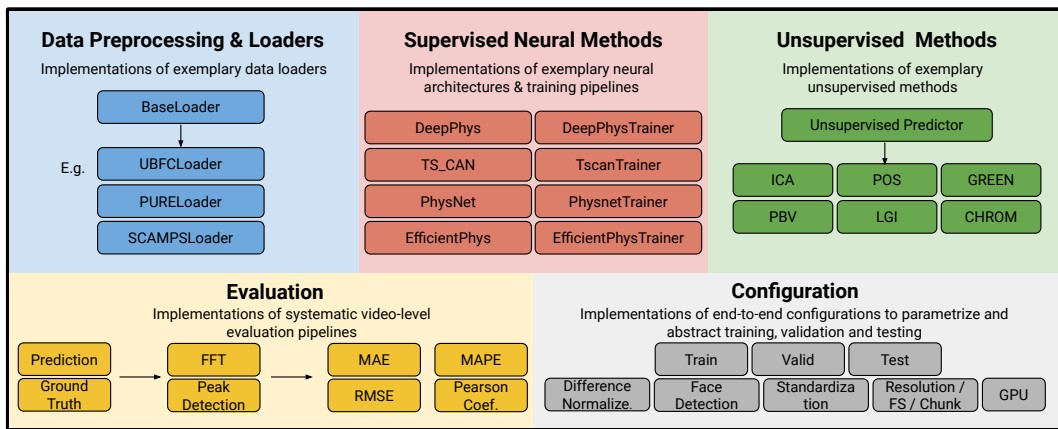

Figure 2: **Overview.** An overview of the rPPG-Toolbox codebase.

difference frames (the difference between each pair of consecutive frames) stored as numpy arrays in a [N, W, H, C] format. Where N is the length of the sequence, W is the width of the frames, H is the height of the frames, and C is the number of channels. There are six channels in this case, as both the raw frames and difference frames account for three color channels each. For faster data loading, all videos in the datasets are typically broken up into several "chunks" of non-overlapping N (e.g., 180) frame sequences. All of these parameters (N, W, H, C) are easy to change and customize. The PPG waveform labels are stored as numpy arrays in a [N, 1] format. The entire pre-processing procedure is supported with multi-thread processing to accelerate the data processing time.

We have provided pre-processing code for UBFC-rPPG [22], PURE [23] SCAMPS [24], MMPD [25], BP4D+ [26], and UBFC-Phys [27]. Each of these datasets encompasses a diverse array of real-world conditions, capturing variations in factors such as motion, lighting, skin tones/types, and backgrounds, thus presenting robust challenges for any signal processing and machine learning algorithm. Tools (python notebooks) are provided for quickly visualizing pre-processed datasets and will be detailed further in Appendix-J. We also support the pre-processing and usage of augmented versions of the UBFC-rPPG [22] and PURE [23] datasets, a feature which we describe further in Section 4.2.

**UBFC-rPPG** [22]: This dataset features RGB videos recorded using a Logitech C920 HD Pro webcam at 30Hz. The videos have a resolution of 640x480, and they are stored in an uncompressed 8-bit RGB format. Reference PPG data was obtained using a CMS50E transmissive pulse oximeter, thereby providing the gold-standard validation data. The subjects were positioned approximately one meter away from the camera during the recording sessions. The videos were captured under indoor conditions with a combination of natural sunlight and artificial illumination.

**PURE** [23]: This dataset consists of recordings from 10 subjects, including 8 males and 2 females. The video footage was captured with an RGB eco274CVGE camera from SVS-Vistek GmbH, with a frequency of 30Hz and a resolution of 640x480. Subjects were positioned approximately 1.1 meters from the camera and were illuminated from the front by ambient natural light filtering through a window. The gold-standard ground truth of PPG and SpO2 were obtained at 60Hz with a CMS50E pulse oximeter affixed to the subject's finger. Each participant completed six recordings under varied motion conditions, thereby contributing to a range of data reflecting different physical states.

**SCAMPS** [24]: This dataset encompasses 2,800 video clips, comprising 1.68M frames, featuring synthetic avatars in alignment with cardiac and respiratory signals. These waveforms and videos were generated by employing a sophisticated facial processing pipeline, resulting in high-fidelity, quasi-photorealistic renderings. To provide robust test conditions, the videos incorporate various confounders such as head motions, facial expressions, and changes in ambient illumination.

**MMPD** [25]: This dataset includes 660 one-minute videos recorded using a Samsung Galaxy S22 Ultra mobile phone, at 30 frames per second with a resolution of 1280x720 pixels and then compressed to 320x240 pixels. The ground truth PPG signals were simultaneously captured using an HKG-07C+ oximeter, at 200 Hz and then downsampled to 30 Hz. It contains Fitzpatrick skin types 3-6, four different lighting conditions (LED-low, LED-high, incandescent, natural), four distinct

activities (stationary, head rotation, talking, and walking), and exercise scenarios. With multiple labels provided, different subsets of this dataset can be easily used for research using our toolbox.

**BP4D+** [26]: This dataset contains video footage captured at a rate of 25 frames per second, for 140 subjects, each participating in 10 emotion-inducing tasks, amounting to a total of 1400 trials and associated videos. In addition to the standard video footage, the dataset also includes 3D mesh models and thermal video, both captured at the same frame rate. Alongside these, the dataset offers supplementary data including blood pressure measurements (wave, systolic, diastolic, mean), heart rate in beats per minute, respiration (wave, rate bpm), electrodermal activity, and Facial Action Coding System (FACS) encodings for specified action units.

**UBFC-Phys** [27]: The UBFC-PHYS dataset, a multi-modal dataset, contains 168 RGB videos, with 56 subjects (46 women and 10 men) per a task. There are three tasks with significant amounts of unconstrained motion under static lighting conditions - a rest task, a speech task, and an arithmetic task. The dataset contains gold-standard blood volume pulse (BVP) and electrodermal activity (EDA) measurements that were collected via the Empatica E4 wristband. The videos were recorded at a resolution of 1024x1024 and 35Hz with a EO-23121C RGB digital camera. We utilized all three tasks and the same subject sub-selection list provided by the authors of the dataset in the second supplementary material of Sabour et al. [27] for evaluation. We reiterate this subject sub-selection list in Appendix-H.

## 3.2 Methods

### 3.2.1 Unsupervised Methods

The following methods all use linear algebra and traditional signal processing to recover the estimated PPG signal: 1) **Green** [28]: the green channel information is used as the proxy for the PPG after spatial averaging of RGB video; 2) **ICA** [29]: Independent Component Analysis (ICA) is applied to normalized, spatially averaged color signals to recover demixing matrices; 3) **CHROM** [30]: a linear combination of the chrominance signals obtained from the RGB video are used for estimation; 4) **POS** [31]: plane-orthogonal-to-the-skin (POS), is a method that calculates a projection plane orthogonal to the skin-tone based on physiological and optical principles. A fixed matrix projection is applied to the spatially normalized, averaged pixel values, which are used to recover the PPG waveform; 5) **PBV** [32]: a signature, that is determined by a given light spectrum and changes of the blood volume pulse, is used in order to derive the PPG waveform while offsetting motion and other noise in RGB videos; 6) **LGI** [33]: a feature representation method that is invariant to motion through differentiable local transformations.

### 3.2.2 Supervised Neural Methods

The following implementations of supervised learning algorithms are included in the toolbox. All implementations were done using PyTorch [37]. Common optimization algorithms, such as Adam [38] and AdamW [39], and criterion, such as mean squared error (MSE) loss, are utilized for training except for where noted. The learning rate scheduler typically follows the 1cycle policy [40], which anneals the learning rate from an initial learning rate to some maximum learning rate and then, from that maximum learning rate, to some learning rate much lower than the initial learning rate. The total steps in this policy are determined by the number of epochs multiplied by the number of training batches in an epoch. The 1cycle policy allows for convergence due to the learning rate being adjusted well below the initial, maximum learning rate throughout the cycle, and after numerous epochs in which the learning rate is much higher than the final learning rate. We found the 1cycle learning rate scheduler to provide stable results with convergence using a maximum learning rate of 0.009 and 30 epochs. We provide parameters in the toolbox that can enable the visualization of the losses and learning rate changes for both the training and validation phases. Further details on these key visualizations for supervised neural methods are provided in the GitHub repository.

**DeepPhys** [4]: A two-branch 2D convolutional attention network architecture. The two representations (appearance and difference frames) are processed by parallel branches with the appearance branch guiding the motion branch via a gated attention mechanism. The target signal is the first differential of the PPG waveform.

**PhysNet** [5]: A 3D convolutional network architecture. Yu et al. compared this 3D-CNN architecture with a 2D-CNN + RNN architecture, finding that a 3D-CNN version was able to achieve superior

Table 2: **Benchmark Results.** Performance on the UBFC-rPPG [22], PURE [23] UBFC-Phys [27] and MMPD [25] datasets generated using the rPPG toolbox. For the supervised methods we show cross-dataset training results using the UBFC-rPPG, PURE and SCAMPS datasets.

| | | | Test Set | | | | | | | |
|---|---|---|---|---|---|---|---|---|---|---|
| | | | PURE [23] | | UBFC-rPPG [22] | | UBFC-Phys [27] | | MMPD [25] | |
| | Method | Train Set | MAE$^\downarrow$ | MAPE$^\downarrow$ | MAE$^\downarrow$ | MAPE$^\downarrow$ | MAE$^\downarrow$ | MAPE$^\downarrow$ | MAE$^\downarrow$ | MAPE$^\downarrow$ |
| UNSUPERVISED | GREEN [28] | N/A | 10.09 | 10.28 | 19.81 | 18.78 | 13.55 | 16.01 | 21.68 | 24.39 |
| | ICA [29] | N/A | 4.77 | 4.47 | 14.70 | 14.34 | 10.03 | 11.85 | 18.60 | 20.88 |
| | CHROM [30] | N/A | 5.77 | 11.52 | 3.98 | 3.78 | 4.49 | 6.00 | 13.66 | 15.99 |
| | LGI [33] | N/A | 4.61 | 4.96 | 15.80 | 14.70 | 6.27 | 7.83 | 17.08 | 18.98 |
| | PBV [32] | N/A | 3.91 | 4.82 | 15.90 | 15.17 | 12.34 | 14.63 | 17.95 | 20.18 |
| | POS [31] | N/A | 3.67 | 7.25 | 4.00 | 3.86 | 4.51 | 6.12 | 12.36 | 14.43 |
| SUPERVISED | TS-CAN [6] | UBFC-rPPG | 3.69 | 3.38 | N/A | N/A | 5.13 | 6.53 | 14.00 | 15.47 |
| | | PURE | N/A | N/A | 1.29 | 1.50 | 5.72 | 7.34 | 13.93 | 15.14 |
| | | SCAMPS | 4.66 | 5.83 | 3.62 | 3.53 | 5.55 | 6.91 | 19.05 | 21.77 |
| | PHYSNET [5] | UBFC-rPPG | 8.06 | 13.67 | N/A | N/A | 5.79 | 7.69 | 9.47 | 11.11 |
| | | PURE | N/A | N/A | 0.98 | 1.12 | 4.78 | 6.15 | 13.93 | 15.61 |
| | | SCAMPS | 13.30 | 20.01 | 5.40 | 5.43 | 8.53 | 11.22 | 20.78 | 24.43 |
| | PHYSFORMER [34] | UBFC-rPPG | 12.92 | 23.92 | N/A | N/A | 6.63 | 8.91 | 12.1 | 15.41 |
| | | PURE | N/A | N/A | 1.44 | 1.66 | 6.04 | 7.67 | 14.57 | 16.73 |
| | | SCAMPS | 26.58 | 42.79 | 4.56 | 5.18 | 11.91 | 15.57 | 22.69 | 27.06 |
| | DEEPPHYS [35] | UBFC-rPPG | 5.54 | 5.32 | N/A | N/A | 6.62 | 8.21 | 17.49 | 19.26 |
| | | PURE | N/A | N/A | 1.21 | 1.42 | 8.42 | 10.18 | 16.92 | 18.54 |
| | | SCAMPS | 3.95 | 4.25 | 3.10 | 3.08 | 4.75 | 5.89 | 15.22 | 16.56 |
| | EFF.PHYS-C [36] | UBFC-rPPG | 5.47 | 5.39 | N/A | N/A | 4.93 | 6.25 | 13.78 | 15.15 |
| | | PURE | N/A | N/A | 2.07 | 2.10 | 5.31 | 6.61 | 14.03 | 15.31 |
| | | SCAMPS | 10.24 | 11.70 | 12.64 | 11.26 | 6.97 | 8.47 | 20.41 | 23.52 |

MAE = Mean Absolute Error in HR estimation (Beats/Min), MAPE = Mean Percentage Error (%).

performance. Therefore, we included the 3D-CNN in this case. Instead of an MSE loss, a negative Pearson loss is utilized. It is worth noting that we used difference-normalized frames as input to PhysNet as the original paper does not specify a concrete input data format. Additional experiments involving raw frame inputs are included in Appendix-D.

**PhysFormer** [34]: PhysFormer is a video transformer-based architecture that adaptively aggregates both local and global spatio-temporal features toward rPPG representation enhancement. The architecture ultimately incorporated and emphasized long-term, global features and allowed for notable improvements in performance relative to various methods, including POS [31], DeepPhys [4], and PhysNet [5]. Instead of an MSE loss, a dynamic loss composed of numerous hyperparameters, a negative Pearson loss, a frequency cross-entropy loss, and a label distribution loss is used. Furthermore, the chosen learning rate scheduler differs from our toolbox's default learning rate scheduler (1cycle policy [40]) in that we effectively opt to use a single, constant learning rate for training. We utilize difference-normalized frames as an input to PhysFormer.

**TS-CAN** [6]: A two-branch 2D convolutional attention network architecture that leverages temporal shift operation information across the time axis to perform efficient temporal and spatial modeling. This network is an on-device, real-time algorithm. The target signal is the first differential of the PPG waveform.

**EfficientPhys-C** [36]: A single-branch 2D convolutional neural network that aims to provide an end-to-end, super lightweight network for real-time on-device computation. The architecture has a normalization module that calculates frame differences and learnable normalization, as well as a self-attention module to help the network focus on skin pixels associated with PPG signal.

## 3.3 Pre-Processing, Training, Post-Processing and Evaluation

In the rPPG-Toolbox, we offer a configuration file system that enables users to modify all parameters used in pre-processing, training, post-processing, and evaluation. A YAML file is provided for every experiment and includes blocks for pre/post-processing, training, validation, testing, model

hyperparameters, and computational resources. The pre/post-processing for neural and unsupervised methods share similar settings, such as the same input resolution and face cropping.

In terms of pre-processing, we provide three input data types: 1) "DiffNormalized", which calculates the difference of every two consecutive frames and labels, and normalizes them by their standard deviation; 2) "Standardized", which standardizes the raw frames and labels using z-score; 3) "Raw", which uses the original frames and labels without modification. Additionally, we provide parameters for face cropping, a vital aspect of our task. In the config file, users can use dynamic detection to perform face cropping every N frames and scale the face bounding box by a coefficient to maintain consistency of face cropping in motion videos. Users can also elect to use a median bounding box with dynamic detection in order to help filter out erroneous detections of the face.

With regard to the training of neural networks, our toolbox provides flexibility to parameterize which portion of the data is used for training, validation, or testing. For instance, we can use the first 80% of UBFC-rPPG for training, the last 20% of UBFC-rPPG for validation and then use the entire PURE dataset for testing. Moreover, the distinct parameters (e.g., dropout rate) of each neural network can be defined in the config file.

For post-processing and evauation, there are several standard post-processing steps that are typically employed to improve model predictions. A 2nd-order Butterworth filter (cut-off frequencies of 0.75 and 2.5 Hz) is applied to filter the predicted PPG waveform. The choice of filtering parameters can have a significant impact on downstream results such as heart rate errors. A Fast Fourier Transform or a peak detection algorithm is then applied to the filtered signal to calculate the heart rate. In this toolbox, we support five metrics for video-level heart rate estimations: mean absolute error (MAE), root mean squared error (RMSE), mean absolute percentage error (MAPE), signal-noise ratio (SNR), and Pearson Correlation ($\rho$), along with a calculation of standard error for a better understanding of the accuracy of the aforementioned metrics. We also give users the option to visualize Bland-Altman plots as a part of evaluation. Finer details on the supported metrics (F), metric results not reported in the main paper (G), and Bland-Altman plots (J.3) appear in the respective appendices. For better reproducibility, we also provide pre-trained models in our Github repository to allow researchers to perform model inference. The detailed definition of each config parameter is also provided in the GitHub repository.

### 3.4 Benchmarking

To show that the implementations of the baseline methods are functioning as expected and to provide benchmark results for consumers of the toolbox to reference and reproduce, we performed a set of baseline experiments using three commonly used video rPPG datasets for training: SCAMPS [24], UBFC-rPPG [22] and PURE [23] and tested on four datasets including UBFC-rPPG [22], PURE [23], UBFC-Phys [27], and MMPD [25]. Except where noted in the GitHub repository, neural models utilized a training batch size of 4, 30 epochs, and an inference batch size of 4 for all experiments. Due to the multi-institution team behind this toolbox, different kinds of GPUs were utilized to produce benchmark results. Tables 2 and 4 were produced on a machine using a single NVIDIA RTX A4500 GPU. Tables 3 and 5 were produced on a machine using a single NVIDIA GeForce 2080 Ti GPU. As illustrated in Table 2, we show MAE and MAPE computed between the video-level heart rate estimations and gold standard measurements. Additional cross-dataset experiment results and metric results can be found in Appendix-G and F.

## 4   Additional Features

### 4.1   Weakly Supervised Training

Supervised rPPG training requires high fidelity synchronous PPG waveform labels. However not all datasets contain such high quality labels. In these cases we offer the option to train on synchronous PPG "pseudo" labels derived through a signal processing methodology as described by [41]. These labels are produced through POS-generated [31] PPG waveforms, which are then bandpass filtered around the normal heart-rate frequencies (cut-off frequencies of 0.70 and 3.0 Hz), and finally amplitude normalized using a Hilbert-signal envelope. The tight filtering and envelope normalization results in a strong periodic proxy signal, but at the cost of limited signal morphology.

For instance, in the BP4D+ dataset [26], the cardiac ground truth is represented by a blood pressure waveform. Although this waveform exhibits the same periodicity as the PPG signal, it has a phase shift that adversely affects model training. A figure that illustrates sample pseudo labels derived for BP4D+ [26] videos plotted against the ground truth blood pressure waveform as shown in Figure 3. Table 3 presents results for supervised methods, trained on BP4D+ [26] pseudo labels. We extend this feature to all of the supported datasets.

Table 3: **Training with Pseudo Labels.** For the supervised methods we show results training with the (entire) BP4D+ [26] dataset, using POS [31] derived pseudo training labels.

| Training Set Testing Set | BP4D+[26] with POS Pseudo Labels | | | | | |
| | UBFC-rPPG [22] | | | PURE [23] | | |
| | MAE↓ | MAPE↓ | $\rho\uparrow$ | MAE↓ | MAPE↓ | $\rho\uparrow$ |
| Supervised | | | | | | |
| TS-CAN [6] | 4.69 | 4.51 | 0.78 | 1.29 | 1.60 | 0.97 |
| PhysNet(Normalized) [5] | 1.78 | 1.92 | 0.96 | 3.69 | 7.35 | 0.88 |
| DeepPhys [35] | 2.74 | 2.81 | 0.93 | 2.47 | 2.49 | 0.89 |
| EfficientPhys-C [36] | 2.43 | 2.52 | 0.90 | 3.59 | 3.27 | 0.80 |

MAE = Mean Absolute Error in HR estimation (Beats/Min), MAPE = Mean Percentage Error (%), $\rho$ = Pearson Correlation in HR estimation.

## 4.2 Motion Augmented Training

The usage of synthetic data in the training of machine learning models for medical applications is becoming a key tool that warrants further research [42]. In addition to providing support for the fully synthetic dataset SCAMPS [24], we provide provide support for synthetic, motion-augmented versions of the UBFC-rPPG [22], PURE [23], SCAMPS [24], and UBFC-PHYS [27] datasets for further exploration toward the use of synthetic data for training rPPG models. The synthetic, motion-augmented datasets are generated using an open-source motion augmentation pipeline targeted for increasing motion diversity in rPPG videos [43]. We present cross-dataset results using a motion-augmented version of the UBFC-rPPG [22] dataset in Table 4. We also provide tools that leverage OpenFace [44] for extracting, visualizing, and analyzing motion in rPPG video datasets. Further details regarding these tools are shared in our GitHub repository.

Table 4: **Training with Motion-Augmented Data.** We demonstrate results training on a motion-augmented (MA) version of the UBFC-rPPG [22] dataset generated using an open-source motion augmentation pipeline [43] and testing on the unaugmented version of the PURE [23] dataset.

| Training Set Testing Set | MAUBFC-rPPG [22] | | | | | | | | |
| | PURE [23] | | | UBFC-Phys [27] | | | MMPD [25] | | |
| | MAE↓ | MAPE↓ | $\rho\uparrow$ | MAE↓ | MAPE↓ | $\rho\uparrow$ | MAE↓ | MAPE↓ | $\rho\uparrow$ |
| TS-CAN [6] | 1.07 | 1.20 | 0.97 | 5.03 | 6.36 | 0.75 | 12.59 | 13.77 | 0.23 |
| PhysNet (Normalized) [5] | 17.03 | 32.37 | 0.38 | 5.51 | 7.50 | 0.68 | 10.67 | 13.99 | 0.33 |
| DeepPhys [35] | 1.15 | 1.40 | 0.97 | 4.95 | 6.26 | 0.75 | 12.71 | 13.70 | 0.21 |
| EfficientPhys-C [36] | 2.59 | 2.67 | 0.88 | 4.80 | 6.10 | 0.79 | 13.39 | 14.50 | 0.14 |

MAE = Mean Absolute Error in HR estimation (Beats/Min), MAPE = Mean Percentage Error (%), $\rho$ = Pearson Correlation in HR estimation.

## 4.3 Extending the rPPG-Toolbox for Physiological Multitasking

While this toolbox is primarily targeted towards rPPG model training and evaluation, it can be easily extended to support multi-tasking of physiological signals. As an example, we implement BigSmall [41], an architecture that multi-tasks PPG, respiration, and facial action. Similar to [41] we present 3-fold cross-validation results across the action unit (AU) subset of BP4D+ [26] (the portion of the dataset with AU labels), and use the same subject folds and hyperparameters as implemented in the original publication. These results can be found in Table 5. Note, that like [41], facial action metrics are calculated across 12 common AUs (AU #s 1, 2, 4, 6, 7, 10, 12, 14, 15, 17, 23, 24). Additional details of training and evaluation could be found in Appendix-I.

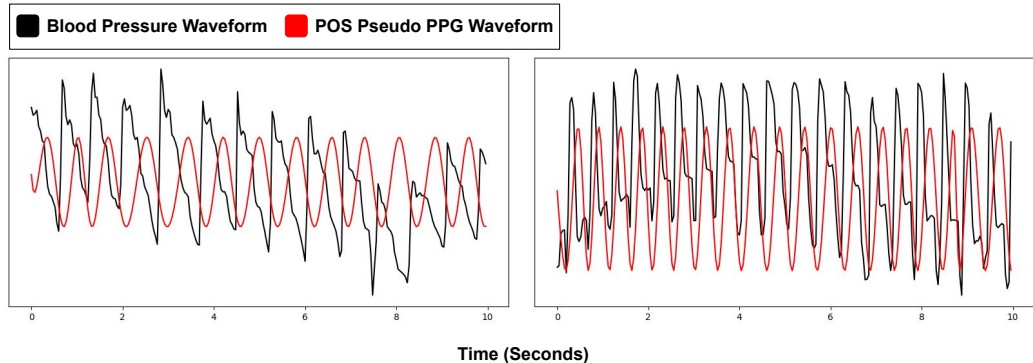

Figure 3: **Generated Pseudo Labels.** Samples of POS [31] generated PPG pseudo labels plotted against ground truth blood pressure waveforms from BP4D+ [26].

Table 5: **Multitasking Results.** For the BigSmall [41] method we show results for multi-tasking PPG, respiration, and action unit classification; training with the BP4D+ [26] (AU subset) dataset, using POS [31] derived pseudo training PPG labels.

| Training Set | | | | | | | | | |
|---|---|---|---|---|---|---|---|---|---|
| Testing Set | | | | BP4D+ [26] | | | | | |
| | | | | BP4D+ [26] | | | | | |
| Task | | rPPG | | | Respiration | | | Facial Action | |
| | MAE↓ | MAPE↓ | $\rho$ ↑ | MAE↓ | MAPE↓ | $\rho$ ↑ | F1↑ | Prec. ↑ | Acc. ↑ |
| BigSmall [41] | 3.23 | 3.51 | 0.83 | 5.19 | 26.28 | 0.14 | 42.82 | 39.85 | 65.73 |

MAE = Mean Absolute Error in HR estimation (Beats/Min), MAPE = Mean Percentage Error (%), $\rho$ = Pearson Correlation in HR estimation, F1 = average F1 across 12 action units, Prec. = average precision across 12 action units, Acc. = average accuracy across 12 action units.

## 4.4 Training, Evaluation and Analysis Features

In this toolbox, we have incorporated a diverse set of training and evaluation functionalities. These include: 1) data pre-processing visualization tools, enabling users to inspect and understand their data before feeding it into the model; 2) comprehensive tracking and visualization of key training metrics such as training loss, validation loss, and learning rate, facilitating a thorough monitoring of the model's learning progress; 3) the implementation of Bland-Altman plots, providing a robust method for assessing agreement between two ground-truth HR and predicted HR; 4) advanced motion analysis capabilities. For an in-depth exploration of these features, we direct the reader to Appendix-J and our Github page, where detailed descriptions and usage examples are provided.

## 5 Limitations

We acknowledge there are many limitations in our current toolbox and plan to continue improve it in the future. In the ensuing phases of our research, we envision a collaborative approach, working in concert with the wider scientific community, to enhance the efficacy and capabilities of the rPPG-Toolbox. The current limitations include 1) this toolbox does not support all of the latest neural architectures and diverse datasets; 2) it does not support unsupervised and self-supervised learning paradigms; 3) it does not support applications beyond of heart rate calculation such as heart rate variability, blood pressure, $SpO_2$, and other importasnt physiological measures.

## 6 Broader Impacts

Camera sensing has advantages and benefits with the potential to make important cardiac measurement more accessible and comfortable. One of the motivating use-cases for rPPG is turning everyday devices equipped with cameras into scalable health sensors. However, pervasive measurement can also feel intrusive. We are releasing the rPPG toolbox with a Responsible AI License [45] that

restricts negative and unintended uses of the toolbox. We also acknowledge the presence of several potential negative concerns and impacts, which are described as follows.

**Privacy Concerns:** Camera-based physiological sensing offers a revolutionary way to extract physiological signals from video recordings, enabling a myriad of applications, from remote patient monitoring to daily fitness tracking. However, these advancements come with significant privacy concerns. First and foremost, the very nature of remote sensing allows for the collection of personal data without direct physical interaction or, in some cases, knowledge of the individual being monitored. This can potentially enable unauthorized entities to capture sensitive physiological data covertly. Furthermore, as these systems become more widespread, there is a risk that everyday places such as shopping malls, public transport, and even workplaces might employ rPPG systems, leading to widespread passive data collection. Such extensive monitoring can lead to privacy invasions, where individuals are constantly under physiological surveillance without their explicit consent.

**Potential Negative Impact:** Beyond individual privacy, there are broader societal implications of widespread contactless camera-based physiological sensing. There's the potential for the creation of a pervasive surveillance state where citizens are continuously monitored, not just for their actions but also for their physiological responses. Such monitoring can lead to "physiological profiling," where individuals are judged, categorized, or even discriminated against based on their bodily responses. For instance, elevated heart rates or other physiological markers might be misinterpreted as signs of nervousness, guilt, or deceit, potentially affecting decision-making in areas such as law enforcement, job interviews, or public services. Moreover, a continuous emphasis on physiological metrics might foster an environment of physiological conformism, where people feel pressured to exhibit 'normal' physiological signs even if they aren't feeling well or are under duress.

**Potential Ethical Concerns:** The ethical implications of rPPG are multi-faceted. Firstly, there is the concern of informed consent. As technology becomes more integrated into our environments, it becomes challenging to ensure that individuals are aware of, and have agreed to, the collection of their physiological data. Moreover, the accuracy and reliability of rPPG systems can vary depending on factors like skin tone, lighting conditions, and other external factors. This introduces the risk of systematic biases, where certain groups might be inaccurately assessed or marginalized due to technological limitations or inherent biases in the algorithms. Ethical concerns also arise from potential misuse. For example, businesses might use rPPG data to gauge consumer reactions to products or advertisements, leading to manipulative strategies that target individual vulnerabilities. In extreme cases, authoritarian regimes might use such technologies to monitor citizens for signs of dissent or unrest. As with any potent tool, the ethical application of rPPG requires careful consideration of its potential for both benevolent and malevolent use.

# 7 Conclusion

Research relies on the sharing of ideas, this not only allows methods to be verified, saving time and resources, but also allows researchers to more effectively build upon existing work. Without these resources and open-sourced code bases, fair evaluation and comparison of methods is difficult, creates needless repetitions, and wastes resources. We present an end-to-end and comprehensive toolbox, called rPPG-Toolbox, containing code for pre-processing multiple public datasets, implementations of supervised machine learning (including training pipeline) and unsupervised methods, and post-processing and evaluation tools.

# 8 Acknowledgement

This project is supported by a Google PhD Fellowship for Xin Liu and a research grant from Cisco for the University of Washington as well as a career start-up funding grant from the Department of Computer Science at UNC Chapel Hill. This research is also supported by Tsinghua University Initiative Scientific Research Program, Beijing Natural Science Foundation, and the Natural Science Foundation of China (NSFC).

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

# A    Overview of Appendices

Our appendices contain the following additional details and results:

- In Section B, we provide details regarding the background of rPPG and an overview of existing methods.
- In Section C, we provide an overview of potential applications out of rPPG technologies.
- In Section D, we provide additional results about PhysNet.
- In Section E, we provide an overview of rPPG network recommendations.
- In Section F we provide details toward metrics supported by our toolbox. We also provide additional metric results in Section G that were not included in the main paper due to space constraints.
- Section H briefly details which subjects we utilized for exclusion, or conversely sub-selection, in each task when dealing with the UBFC-Phys [27] dataset. We also briefly describe video filtering criteria available via the toolbox and useful for subject sub-selection.
- Additional details related to training and evaluation for physiolgogical multitasking is shared in Section I.
- Section J briefly describes additional features included in the toolbox. These features, including pre-processed data visualization, loss and learning visualization, Bland-Altman plots, and motion analysis, are further detailed with exemplar usage in the rPPG-Toolbox's GitHub repo: `https://github.com/ubicomplab/rPPG-Toolbox`

# B    Background and Existing Research in rPPG

## B.1    Background

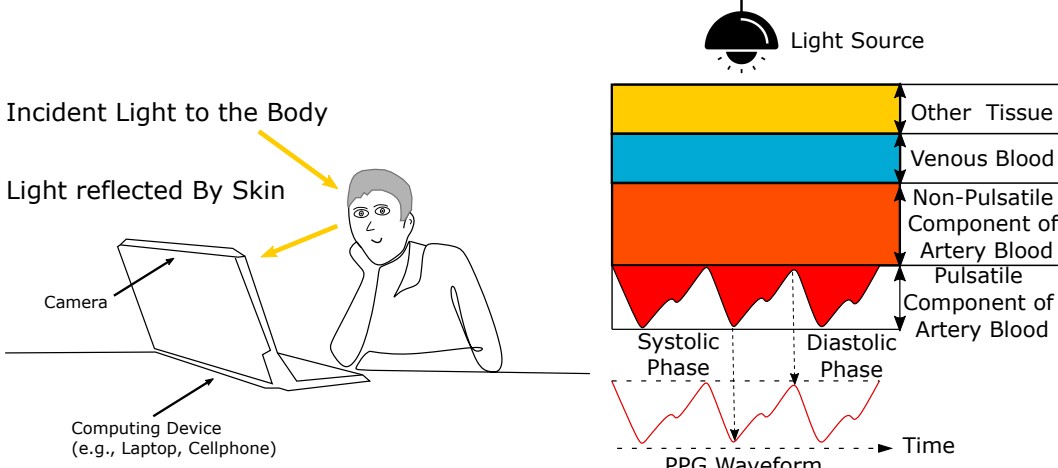

Figure 4: The principles behind camera-based physiological sensing. The volumetric changes of blood under the surface of the skin cause changes in light absorption and reflection, which is the source of PPG signal.

Remote PPG (rPPG) measurement involves the development of computational methods for extracting physiological parameters (e.g., pulse rate, respiration rate, blood oxygenation, blood pressure) based on light reflected from, or transmitted through, the human body. Essentially these methods use pixel information to quantify changes in visible light, or other electromagnetic radiation (e.g., infrared or thermal), that are modulated by blood flow in the periphery of the skin (see Fig.4). This reflected radiation is also affected by body motions and absorption characteristics of the skin [46, 47, 48, 49, 50, 51, 31]. In this article, we will focus primarily on the use of visible light, due to the ubiquitous nature of RGB cameras.

As visible light penetrates between 4 to 5 mm below the skin's surface, it is modulated by the volume of oxygenated and deoxygenated hemoglobin enabling the measurement of the peripheral blood

volume pulse (BVP). The frequency channels offered by multiband (e.g., RGB) cameras enable the composition of blood, including the oxygen saturation to be measured. In addition, these pixels are affected by the motion as a person breaths in and out and by the mechanical effects of the heart beating, enabling the measurement of breathing signals and the ballistocardiogram (BCG). Analyzing the morphology of these signals, and combining them together, offers the possibility of measuring correlates of blood pressure.

## B.2 Optical Principals

The Lambert-Beer law (LBL) [50, 51, 35] and Shafer's dichromatic reflection model (DRM) [31] are two models which provide a framework for capturing the effects of an imager, lighting, body motions and physiological processes on recorded pixel intensities. Given the optical characteristics of oxygenated and deoxygenated blood, we also have priors on the wavelengths of light that contain the strongest or weakest pulsatile information. This prior knowledge is important for measuring physiological parameters accurately. Most computational methods are built upon this grounding.

## B.3 Algorithms

Many computational approaches for recovering physiological signals from videos have similar steps. The first step typically involves localizing a region of interest within each video frame. In a large majority of cases the face or head are the region of interest and therefore facial detection and/or landmark detection are used. However, in other cases skin segmentation might be preferred. Aggregating pixels spatially is a subsequent step that has been used to help to reduce noise from camera quantization errors. The operation can be performed by downsampling an image [35] or simply averaging all pixel values with a region of interest [49]. Many cameras capture frames from more than one frequency band (e.g., RGB) that provide complementary measurements to capture different properties of the light reflected from the body. This information can be used in two ways: 1) for understanding the composition of the blood (e.g., oxygen saturation), 2) improving the signal-to-noise ratio of the recovered blood volume pulse. Typically, computational methods leverage multiple bands and learn a linear or non-linear signal decomposition to estimate the pulse waveform. This manipulation of the color channel signals can be grounded in the optical properties of the skin [31, 30] or learned in a data-driven manner given a specialized learning criteria or loss function.

More recently supervised machine learning has become the most popular approach. Specifically, deep learning and convolutional neural networks provide the current state-of-the-art results. These methods present the opportunity for more "end-to-end" learning and researchers have gradually tried to replace handcrafted processing steps with learnable components. Since the relationship between underlying physiological signal and skin pixels in a video is complicated, deep neural networks have shown superior performance on modeling such non-linear relationship compared to traditional source separation methods [35, 5, 52, 53, 6, 54, 55, 56, 57, 34]. Moreover, due to the flexibility of neural network, researchers have also explored neural based methods for real-time on-device inference [6, 36, 58, 59, 60, 61], self-supervised learning [14, 53, 62, 15, 13, 16], domain generalization [11, 63, 64] and estimating new vital measurement such as blood pressure [65, 66].

## C Potential Applications of rPPG

The SARS-CoV-2 (COVID-19) pandemic has accelerated the pace of change in healthcare services. In particular, how healthcare services are delivered around the world has needed to be rethought in the presence of new risks to patients and providers and restrictions on travel. The virus has been linked to the increased risk of cardiopulmonary (heart and lung related) illness with symptoms such as respiratory distress syndrome, myocarditis, and the associated chronic damage to the cardiovascular system. Experts suggest that particular attention should be given to cardiovascular protection during treatment of COVID-19 [67]. While measurement is not the sole solution to these problems, they have acutely highlighted the need for scalable and accurate physiological monitoring. Ubiquitous or pervasive health sensing technology could help patients conduct daily screenings, monitor the effects of medication on their symptoms, and help clinicians make more informed and accurate decisions.

The potential advantages that video-based contactless measurement offers have helped to draw a significant amount of attention to the field in recent years. Contact biomedical sensors (e.g.,

electrocardiograms, pulse oximeters) are the standard used for clinical screening and at-home measurement. However, these devices are usually bulky and are still not ubiquitously available, especially in low-resource settings. On the other hand, non-contact camera-based physiological sensing presents a new opportunity for highly scalable and low-cost physiological monitoring through ordinary cameras (e.g., webcams or smartphone cameras) [29]. Besides the convenience and potential scalability, this technology could also reduce the risk of infection for vulnerable patients and discomfort caused by obtrusive leads and electrodes [68]. Finally, we believe there are two specifically compelling advantages of cameras over contact sensors. The first, is that they can capture multi-modal signals, including but not limited to, the activity of the subject, their appearance, facial expressions and gestures, motor control and context. One reason this helps is that the physiological measurements can be interpreted in context. For example, if someone appears in pain, an elevated heart rate can be interpreted differently than without in pain. Secondly, cameras are spatial sensors allowing for the measurement of signals from multiple parts of the body to be measured concomitantly, presenting greater opportunities for characterizing vascular parameters such as pulse transit time.

We would also argue that camera-based physiological sensing could be an influential technology in telehealth. Current telehealth procedures are mainly telephone or video-based communication services where patients see their physician or healthcare provider via Cisco Webex, Zoom or Microsoft Teams. Performing clinical visits at home increases the efficiency of clinical visits and helps people who live in remote locations. There is still a debate over whether high-quality care can be delivered over telehealth platforms. One of notable issues with current telehealth systems is that there is no way for physicians to assess patient's physiological states. The development of accurate and efficient non-contact camera-based physiological sensing technology would provide remote physicians access to the physiological data to make more informed clinical decisions.

# D  Investigation of PhysNet

Table 6: **PhysNet Ablation Results.** Investigation of variants of PhysNet models on the PURE [23] and UBFC-rPPG [22] datasets obtained using the rPPG toolbox.

| | | Test Set | | | |
| | | PURE [23] | | UBFC-rPPG [22] | |
| **Method** | **Train Set** | MAE$^{\downarrow}$ | MAPE$^{\downarrow}$ | MAE$^{\downarrow}$ | MAPE$^{\downarrow}$ |
|---|---|---|---|---|---|
| | UBFC-rPPG | 19.25 | 33.75 | N/A | N/A |
| PHYSNET-RAW [5] | PURE | N/A | N/A | 11.17 | 11.64 |
| | SCAMPS | 18.40 | 31.74 | 10.57 | 11.04 |
| | UBFC-rPPG | 8.06 | 13.67 | N/A | N/A |
| PHYSNET-DIFFNORM [6] | PURE | N/A | N/A | 0.98 | 1.12 |
| | SCAMPS | 13.30 | 20.10 | 5.40 | 5.42 |
| | UBFC-rPPG | 4.80 | 8.46 | N/A | N/A |
| PHYSNET-RAW-TUNED [35] | PURE | N/A | N/A | 1.99 | 1.86 |
| | SCAMPS | 14.35 | 23.00 | 10.33 | 10.71 |

Metrics explained: MAE = Mean Absolute Error in HR estimation (Beats/Min), MAPE = Mean Percentage Error (%).

Based on the rPPG community's help and suggestions, we found that there are two important implementation details missed in the original PhysNet paper: 1) the raw input range has to be set to 0-1; 2) adding normalization to the model output. However, with these changes and the raw input format, PhysNet is not able to achieve results that are close to SOTA in UBFC and PURE. As shown in Table 6's PhysNet-Raw-Tuned, we further fine-tuned the training parameters (changed learning rate from 0.009 to 0.09) and post-processing bandpass filter parameters (set bandpass filtering parameters to [0.5,2.5], detrend value to 200). However, we don't recommend using these network specified parameters as it is not a fair comparison across different network architectures. This toolbox aims to provide a standardized training regime across all the networks and datasets. All in all, we recommend using DiffNorm frames as the input to PhysNet to make training easier to converge.

We also notice that results while training on SCMAPS and testing on PURE are not good. It is worth noting that SCAMPS is a synthetic dataset and can easily cause overfitting with a complex network architecture. Unlike other 2D-CNN based networks, PhysNet is a 3D-CNN based network which is easier to overfit simple datasets.

# E Network Recommendation

In the dynamic realm of rPPG research, architectural choices critically influence the balance between efficiency and accuracy. This section elucidates our findings and offers recommendations on architectures, factoring both computational aspects and performance nuances.

- **Mobile and Computational Efficiency**: For scenarios demanding computational frugality, such as mobile or edge deployments, 2D-CNN based architectures are recommended. They gracefully balance performance and computational overhead, ensuring expeditious responses in resource-limited environments.

- **High-Performance Scenarios**: 3D-CNN or Transformer-based networks, while resource-intensive, offer superior performance. These architectures are apt for applications with lenient resource constraints where the pinnacle of accuracy is sought.

- **Performance Saturation**: Notably, a saturation trend was observed across several network architectures. This insinuates diminishing performance returns with increased complexity or depth, emphasizing the need to pragmatically select architectures based on actual task necessities.

A salient lesson from our research emphasizes the indispensability of diverse dataset incorporation. The rPPG community grapples with challenges such as: 1) motion-infused videos, introducing substantial rPPG signal noise; 2) videos under different lighting conditions (e.g., natural light, LED, incandescent), affecting visual cues critical for models; 3) featuring darker-skinned individuals, historically underrepresented, leading to potential model biases. To bolster robustness and cater to diverse real-world scenarios, it's imperative to amalgamate videos encapsulating the aforementioned conditions in training sets. Such holistic training paradigms ensure the model's aptitude in handling a vast spectrum of real-world challenges.

# F Metric Details

## F.1 rPPG Metrics

We present explanations of metrics supported by our toolbox below.

**Mean Absolute Error (MAE)**: For predicted signal rate $R_p$, ground truth signal rate $R_g$, and for $N$ instances:

$$MAE = \frac{1}{N} \sum_{n=1}^{N} |R_g - R_p|$$

**Root Mean Square Error (RMSE)**: For predicted signal rate $R_p$, ground truth signal rate $R_g$, and for $N$ instances:

$$RMSE = \sqrt{\frac{1}{N} \sum_{n=1}^{N} (R_g - R_p)^2}$$

**Mean Absolute Percentage Error (MAPE)**: For predicted signal rate $R_p$, ground truth signal rate $R_g$, and for $N$ instances:

$$MAPE = \frac{1}{N} \sum_{n=1}^{N} \left| \frac{R_g - R_p}{R_g} \right|$$

**Pearson Correlation ($\rho$)**: For predicted signal rate $R_p$, ground truth signal rate $R_g$, and for $N$ instances, and $\overline{R}$ the average of $R$ for $N$ samples:

$$\rho = \frac{\sum_{n=1}^{N} \left( R_{g.n} - \overline{R_g} \right) \left( R_{p.n} - \overline{R_p} \right)}{\sqrt{\left( \sum_{n=1}^{N} R_{g.n} - \overline{R_g} \right)^2 \left( \sum_{n=1}^{N} R_{p.n} - \overline{R_p} \right)^2}}$$

**Signal-to-Noise Ratio (SNR)**: As in [30], we calculate the Signal-to-Noise Ratio (SNR) for a predicted signal as the ratio between the area under the curve of the power spectrum around the first and second harmonic of the ground truth heart rate frequency and the area under the curve of the rest of the power spectrum. This is mathematically represented as follows:

$$SNR = 10log_{10}\left(\frac{\sum_{45}^{150}(U_t(f)S(f))^2}{\sum_{45}^{150}((1-U_t(f))S(f))^2}\right)$$

Where $S$ is the power spectrum of the estimated rPPG signal. $U_t(f)$ is equal to 1 around the first and second harmonics of the ground truth rPPG signal, while being 0 elsewhere in the power spectrum. In the context of the rPPG-Toolbox, only the power spectrum between 0.75 Hz and 2.5 Hz, or 45 beats/min and 150 beats/min, is considered. We report the mean of the SNR values calculated per video or test sample, such that:

$$MSNR = \frac{1}{N}\sum_{n=1}^{N} SNR$$

**Standard Error ($\pm$ SE)**: The standard error is a measure of the statistical accuracy of an estimate, such as the mean, and is equal to the standard deviation of the theoretical distribution of a large population of such estimates. The standard error takes into account the number of samples utilized in measurement, which is especially useful in the case of remote PPG datasets where the number of test samples can vary significantly from dataset to dataset. For all metrics except for the Pearson correlation ($\rho$), we calculate the standard error as:

$$SE = \frac{\sigma}{\sqrt{n}}$$

Where $\sigma$ is the standard deviation and $n$ is the number of samples. For the Pearson correlation ($\rho$), the standard error is calculated as:

$$SE_\rho = \sqrt{\frac{1-r^2}{n-2}}$$

Where $r$ is the correlation coefficient and $n$ is the number of samples. Similar to how a standard deviation is reported, we report standard error as $\pm SE$.

### F.2 Additional Multitask Metrics

We present explanations of additional metrics added to evaluate the BigSmall [41] model in order to exemplify how this toolbox can be extended to support physiological multitasking.

**Evaluated Action Units (AU)**: Similar to [41], and other AU literature, facial action metrics are calculated for the following 12 commonly used AUs: AU01, AU02, AU04, AU06, AU07, AU10, AU12, AU14, AU15, AU17, AU23, AU24.

**F1**: The harmonic mean of recall and precision. For true positive count $TP$, false positive count $FP$, and false negative count $FN$.

$$F1 = 100 * \frac{2TP}{2TP + FP + FN}$$

**Precision (Prec.)**: For true positive count $TP$, and false positive count $FP$:

$$Precision = 100 * \frac{TP}{TP + FP}$$

**Accuracy (Acc.)** For true positive count $TP$, true negative count $TN$, false positive count $FP$, and false negative count $FN$:

$$Accuracy = 100 * \frac{TP + TN}{TP + TN + FP + FN}$$

# G   Additional Results

We reiterate results provided in the main paper and present additional results including the RMSE, SNR, Pearson correlation, and the corresponding standard errors. Note that there may be minor differences between results in the following tables and the main paper, as they were generated on a different machine using the latest version of the rPPG-Toolbox.

Table 7: **Benchmark Results.** Performance on the PURE [23] and UBFC-rPPG [22] datasets obtained using the rPPG toolbox. For the supervised methods we show cross-dataset training results using the UBFC-rPPG, PURE, and SCAMPS datasets.

| | Method | Train Set | MAE↓ | RMSE↓ | MAPE↓ | $\rho\uparrow$ | SNR↑ |
|---|---|---|---|---|---|---|---|
| | | | \multicolumn{5}{Test Set PURE [23]} | | | | |
| UNSUPERVISED | GREEN [28] | N/A | 10.09 ± 2.81 | 23.85 ± 217.81 | 10.28 ± 2.33 | 0.34 ± 0.12 | -2.66 ± 1.43 |
| | ICA [29] | N/A | 4.77 ± 2.08 | 16.07 ± 153.84 | 4.47 ± 1.65 | 0.72 ± 0.09 | 5.24 ± 1.77 |
| | CHROM [30] | N/A | 5.77 ± 1.79 | 14.93 ± 81.53 | 11.52 ± 3.75 | 0.81 ± 0.08 | 4.58 ± 0.85 |
| | LGI [33] | N/A | 4.61 ± 1.91 | 15.38 ± 134.14 | 4.96 ± 1.72 | 0.77 ± 0.08 | 4.50 ± 1.21 |
| | PBV [32] | N/A | 3.92 ± 1.61 | 12.99 ± 123.60 | 4.84 ± 1.49 | 0.84 ± 0.07 | 2.30 ± 1.31 |
| | POS [31] | N/A | 3.67 ± 1.46 | 11.82 ± 66.87 | 7.25 ± 3.03 | 0.88 ± 0.06 | 6.87 ± 0.95 |
| SUPERVISED | TS-CAN [6] | UBFC-rPPG | 3.69 ± 1.74 | 13.8 ± 113.84 | 3.39 ± 1.44 | 0.82 ± 0.08 | 5.26 ± 1.11 |
| | | SCAMPS | 4.66 ± 1.68 | 13.69 ± 92.53 | 5.83 ± 2.03 | 0.82 ± 0.08 | 0.95 ± 1.04 |
| | PHYSNET [5] | UBFC-rPPG | 8.06 ± 2.34 | 19.71 ± 129.34 | 13.67 ± 4.04 | 0.61 ± 0.11 | 6.68 ± 1.16 |
| | | SCAMPS | 13.30 ± 2.00 | 20.30 ± 94.85 | 20.01 ± 2.97 | 0.51 ± 0.11 | -8.73 ± 0.78 |
| | PHYSFORMER [34] | UBFC-rPPG | 12.92 ± 2.69 | 24.36 ± 132.24 | 23.92 ± 5.22 | 0.47 ± 0.12 | 2.16 ± 1.05 |
| | | SCAMPS | 26.58 ± 2.14 | 31.24 ± 133.33 | 42.79 ± 4.06 | 0.12 ± 0.13 | -12.56 ± 0.53 |
| | DEEPPHYS [35] | UBFC-rPPG | 5.54 ± 2.30 | 18.51 ± 173.09 | 5.32 ± 1.90 | 0.66 ± 0.10 | 4.40 ± 1.32 |
| | | SCAMPS | 3.96 ± 1.67 | 13.44 ± 98.86 | 4.25 ± 1.60 | 0.83 ± 0.07 | 5.07 ± 1.15 |
| | EFF.PHYS-C [36] | UBFC-rPPG | 5.47 ± 2.10 | 17.04 ± 143.80 | 5.40 ± 1.76 | 0.71 ± 0.09 | 4.09 ± 1.16 |
| | | SCAMPS | 10.24 ± 2.48 | 21.65 ± 173.96 | 11.70 ± 2.28 | 0.46 ± 0.12 | -5.49 ± 1.05 |

| | Method | Train Set | MAE↓ | RMSE↓ | MAPE↓ | $\rho\uparrow$ | SNR↑ |
|---|---|---|---|---|---|---|---|
| | | | \multicolumn{5}{Test Set UBFC-rPPG [22]} | | | | |
| UNSUPERVISED | GREEN [28] | N/A | 19.73 ± 3.75 | 31.00 ± 235.38 | 18.72 ± 3.33 | 0.37 ± 0.15 | -11.18 ± 1.63 |
| | ICA [29] | N/A | 16.00 ± 3.09 | 25.65 ± 163.58 | 15.35 ± 2.77 | 0.44 ± 0.14 | -9.91 ± 1.78 |
| | CHROM [30] | N/A | 4.06 ± 1.21 | 8.83 ± 33.93 | 3.84 ± 1.10 | 0.89 ± 0.07 | -2.96 ± 1.18 |
| | LGI [33] | N/A | 15.80 ± 3.67 | 28.55 ± 236.17 | 14.70 ± 3.20 | 0.36 ± 0.15 | -8.15 ± 1.41 |
| | PBV [32] | N/A | 15.90 ± 3.25 | 26.40 ± 199.71 | 15.17 ± 2.91 | 0.48 ± 0.14 | -9.16 ± 1.35 |
| | POS [31] | N/A | 4.08 ± 1.01 | 7.72 ± 21.87 | 3.93 ± 0.91 | 0.92 ± 0.06 | -2.39 ± 1.14 |
| SUPERVISED | TS-CAN [6] | PURE | 1.30 ± 0.40 | 2.87 ± 3.05 | 1.50 ± 0.47 | 0.99 ± 0.02 | 1.49 ± 1.13 |
| | | SCAMPS | 3.62 ± 0.91 | 6.92 ± 18.30 | 3.53 ± 0.84 | 0.93 ± 0.06 | -3.91 ± 0.98 |
| | PHYSNET [5] | PURE | 0.98 ± 0.35 | 2.48 ± 2.55 | 1.12 ± 0.42 | 0.99 ± 0.02 | 1.09 ± 1.15 |
| | | SCAMPS | 5.40 ± 1.46 | 10.89 ± 48.53 | 5.43 ± 1.38 | 0.82 ± 0.09 | -4.97 ± 1.03 |
| | PHYSFORMER [34] | PURE | 1.44 ± 0.54 | 3.77 ± 7.93 | 1.66 ± 0.62 | 0.98 ± 0.03 | 0.18 ± 1.12 |
| | | SCAMPS | 4.56 ± 1.46 | 10.48 ± 68.96 | 5.18 ± 1.93 | 0.81 ± 0.09 | -6.34 ± 0.80 |
| | DEEPPHYS [35] | PURE | 1.21 ± 0.41 | 2.90 ± 3.75 | 1.42 ± 0.49 | 0.99 ± 0.02 | 1.74 ± 1.16 |
| | | SCAMPS | 3.10 ± 1.44 | 9.81 ± 74.70 | 3.08 ± 1.32 | 0.87 ± 0.08 | -0.79 ± 1.22 |
| | EFF.PHYS-C [36] | PURE | 2.07 ± 0.92 | 6.32 ± 32.01 | 2.10 ± 0.87 | 0.94 ± 0.05 | -0.12 ± 1.20 |
| | | SCAMPS | 12.64 ± 3.15 | 23.99 ± 182.44 | 11.26 ± 2.67 | 0.34 ± 0.15 | -9.36 ± 1.05 |

MAE = Mean Absolute Error in HR estimation (Beats/Min), RMSE = Root Mean Square Error in HR estimation (Beats/Min), MAPE = Mean Percentage Error (%), $\rho$ = Pearson Correlation in HR estimation, SNR = Signal-to-Noise Ratio (dB) when comparing predicted spectrum to ground truth spectrum.

Table 8: **Benchmark Results.** Performance on the UBFC-Phys [27] and MMPD [25] datasets generated using the rPPG toolbox. For the supervised methods we show cross-dataset training results using the UBFC-rPPG, PURE and SCAMPS datasets.

**Test Set — UBFC-Phys [27]**

| | Method | Train Set | MAE↓ | RMSE↓ | MAPE↓ | $\rho$↑ | SNR↑ |
|---|---|---|---|---|---|---|---|
| UNSUPERVISED | GREEN [28] | N/A | 13.55 ± 1.30 | 18.80 ± 48.87 | 16.01 ± 1.42 | 0.29 ± 0.10 | -10.34 ± 0.65 |
| | ICA [29] | N/A | 10.04 ± 1.20 | 15.73 ± 43.63 | 11.85 ± 1.35 | 0.36 ± 0.09 | -5.28 ± 0.98 |
| | CHROM [30] | N/A | 4.49 ± 0.60 | 7.56 ± 13.84 | 6.00 ± 0.88 | 0.80 ± 0.06 | -1.92 ± 0.85 |
| | LGI [33] | N/A | 6.27 ± 0.83 | 10.41 ± 22.76 | 7.83 ± 0.99 | 0.70 ± 0.07 | -3.30 ± 0.91 |
| | PBV [32] | N/A | 12.34 ± 1.22 | 17.43 ± 47.24 | 14.63 ± 1.33 | 0.33 ± 0.09 | -9.33 ± 0.71 |
| | POS [31] | N/A | 4.51 ± 0.68 | 8.16 ± 17.36 | 6.12 ± 0.99 | 0.77 ± 0.06 | -1.28 ± 0.90 |
| SUPERVISED | TS-CAN [6] | UBFC-rPPG | 5.13 ± 0.63 | 8.12 ± 18.47 | 6.53 ± 0.85 | 0.76 ± 0.07 | -1.95 ± 0.81 |
| | | PURE | 5.72 ± 0.66 | 8.78 ± 16.94 | 7.34 ± 0.90 | 0.72 ± 0.07 | -3.72 ± 0.78 |
| | | SCAMPS | 5.55 ± 0.67 | 8.71 ± 16.96 | 6.91 ± 0.85 | 0.72 ± 0.07 | -4.40 ± 0.66 |
| | PHYSNET [5] | UBFC-rPPG | 5.79 ± 0.76 | 9.60 ± 17.64 | 7.69 ± 1.07 | 0.70 ± 0.07 | -1.63 ± 0.99 |
| | | PURE | 4.78 ± 0.72 | 8.68 ± 18.99 | 6.15 ± 0.98 | 0.73 ± 0.07 | -0.71 ± 1.00 |
| | | SCAMPS | 8.53 ± 0.98 | 13.02 ± 33.08 | 11.22 ± 1.35 | 0.43 ± 0.10 | -7.15 ± 0.60 |
| | PHYSFORMER [34] | UBFC-rPPG | 6.63 ± 0.77 | 10.22 ± 18.12 | 8.91 ± 1.12 | 0.69 ± 0.07 | -3.58 ± 0.93 |
| | | PURE | 6.04 ± 0.76 | 9.77 ± 18.38 | 7.67 ± 0.99 | 0.65 ± 0.08 | -2.16 ± 0.95 |
| | | SCAMPS | 11.91 ± 1.13 | 16.42 ± 46.43 | 15.57 ± 1.64 | 0.27 ± 0.097 | -10.38 ± 0.39 |
| | DEEPPHYS [35] | UBFC-rPPG | 6.62 ± 0.84 | 10.69 ± 25.90 | 8.21 ± 1.04 | 0.66 ± 0.08 | -2.98 ± 0.82 |
| | | PURE | 8.42 ± 1.09 | 13.80 ± 38.06 | 10.18 ± 1.29 | 0.44 ± 0.09 | -4.41 ± 0.84 |
| | | SCAMPS | 4.75 ± 0.58 | 7.50 ± 14.47 | 5.89 ± 0.72 | 0.82 ± 0.06 | -2.04 ± 0.76 |
| | EFF.PHYS-C [36] | UBFC-rPPG | 4.93 ± 0.58 | 7.65 ± 14.44 | 6.25 ± 0.79 | 0.79 ± 0.06 | -2.09 ± 0.82 |
| | | PURE | 5.31 ± 0.78 | 9.44 ± 27.67 | 6.61 ± 0.96 | 0.70 ± 0.07 | -2.22 ± 0.81 |
| | | SCAMPS | 6.97 ± 0.79 | 10.58 ± 22.70 | 8.47 ± 0.91 | 0.64 ± 0.08 | -7.38 ± 0.47 |

**Test Set — MMPD [25]**

| | Method | Train Set | MAE↓ | RMSE↓ | MAPE↓ | $\rho$↑ | SNR↑ |
|---|---|---|---|---|---|---|---|
| UNSUPERVISED | GREEN [28] | N/A | 21.68 ± 0.67 | 27.69 ± 42.21 | 24.39 ± 0.64 | -0.01 ± 0.04 | -14.34 ± 0.26 |
| | ICA [29] | N/A | 18.60 ± 0.61 | 24.30 ± 33.80 | 20.88 ± 0.58 | 0.01 ± 0.04 | -13.84 ± 0.27 |
| | CHROM [30] | N/A | 13.66 ± 0.50 | 18.76 ± 23.82 | 16.00 ± 0.57 | 0.08 ± 0.04 | -11.74 ± 0.21 |
| | LGI [33] | N/A | 17.08 ± 0.62 | 23.32 ± 34.46 | 18.98 ± 0.60 | 0.04 ± 0.04 | -13.15 ± 0.25 |
| | PBV [32] | N/A | 17.95 ± 0.60 | 23.58 ± 32.45 | 20.18 ± 0.58 | 0.09 ± 0.04 | -13.88 ± 0.24 |
| | POS [31] | N/A | 12.36 ± 0.49 | 17.71 ± 23.65 | 14.43 ± 0.55 | 0.18 ± 0.04 | -11.53 ± 0.22 |
| SUPERVISED | TS-CAN [6] | UBFC-rPPG | 14.01 ± 0.61 | 21.04 ± 30.02 | 15.48 ± 0.61 | 0.24 ± 0.04 | -10.18 ± 0.28 |
| | | PURE | 13.94 ± 0.64 | 21.61 ± 33.02 | 15.15 ± 0.63 | 0.20 ± 0.04 | -9.94 ± 0.27 |
| | | SCAMPS | 19.05 ± 0.58 | 24.20 ± 31.90 | 21.77 ± 0.60 | 0.14 ± 0.04 | -13.24 ± 0.25 |
| | PHYSNET [5] | UBFC-rPPG | 9.47 ± 0.50 | 16.01 ± 22.74 | 11.11 ± 0.58 | 0.31 ± 0.04 | -8.15 ± 0.26 |
| | | PURE | 13.93 ± 0.57 | 20.29 ± 27.57 | 15.61 ± 0.59 | 0.17 ± 0.04 | -10.59 ± 0.27 |
| | | SCAMPS | 20.78 ± 0.55 | 25.09 ± 31.92 | 24.43 ± 0.62 | 0.17 ± 0.04 | -15.86 ± 0.20 |
| | PHYSFORMER [34] | UBFC-rPPG | 12.1 ± 0.51 | 17.79 ± 23.77 | 15.41 ± 0.74 | 0.17 ± 0.04 | -10.53 ± 0.20 |
| | | PURE | 14.57 ± 0.57 | 20.71 ± 29.1 | 16.73 ± 0.63 | 0.15 ± 0.039 | -12.15 ± 0.22 |
| | | SCAMPS | 22.69 ± 0.57 | 26.94 ± 34.41 | 27.06 ± 0.70 | 0.15 ± 0.04 | -18.83 ± 0.32 |
| | DEEPPHYS [35] | UBFC-rPPG | 17.50 ± 0.70 | 25.00 ± 38.62 | 19.27 ± 0.68 | 0.06 ± 0.04 | -11.72 ± 0.33 |
| | | PURE | 16.92 ± 0.70 | 24.61 ± 38.03 | 18.54 ± 0.68 | 0.05 ± 0.04 | -11.53 ± 0.31 |
| | | SCAMPS | 15.22 ± 0.68 | 23.17 ± 38.46 | 16.56 ± 0.66 | 0.09 ± 0.04 | -10.23 ± 0.31 |
| | EFF.PHYS-C [36] | UBFC-rPPG | 13.78 ± 0.68 | 22.25 ± 37.94 | 15.15 ± 0.70 | 0.09 ± 0.04 | -9.13 ± 0.31 |
| | | PURE | 14.03 ± 0.64 | 21.62 ± 32.95 | 15.32 ± 0.63 | 0.17 ± 0.04 | -9.95 ± 0.29 |
| | | SCAMPS | 20.41 ± 0.57 | 25.06 ± 31.72 | 23.52 ± 0.61 | 0.11 ± 0.04 | -14.28 ± 0.24 |

MAE = Mean Absolute Error in HR estimation (Beats/Min), RMSE = Root Mean Square Error in HR estimation (Beats/Min), MAPE = Mean Percentage Error (%), $\rho$ = Pearson Correlation in HR estimation, SNR = Signal-to-Noise Ratio (dB) when comparing predicted spectrum to ground truth spectrum.

Table 9: **Training with Motion-Augmented Data.** We demonstrate results training on a motion-augmented (MA) version of the UBFC-rPPG [22] dataset generated using an open-source motion augmentation pipeline [43] and testing on the unaugmented versions of the PURE [23] dataset, UBFC-Phys [27], and MMPD [25] datasets.

| Training Set | MAUBFC-rPPG [22] | | | | |
| Testing Set | PURE [23] | | | | |
| Metric (± Std. Err.) | MAE↓ | RMSE ↓ | MAPE↓ | $\rho$ ↑ | SNR ↑ |
| --- | --- | --- | --- | --- | --- |
| **Supervised** | | | | | |
| TS-CAN [6] | $1.07 \pm 0.75$ | $5.89 \pm 33.75$ | $1.20 \pm 0.83$ | $0.97 \pm 0.03$ | $8.86 \pm 0.95$ |
| PhysNet (Normalized) [5] | $17.03 \pm 2.97$ | $28.50 \pm 149.16$ | $32.37 \pm 5.82$ | $0.38 \pm 0.12$ | $7.27 \pm 0.88$ |
| DeepPhys [35] | $1.15 \pm 0.76$ | $5.95 \pm 33.75$ | $1.40 \pm 0.85$ | $0.97 \pm 0.03$ | $9.94 \pm 1.00$ |
| EfficientPhys-C [36] | $2.59 \pm 1.43$ | $11.29 \pm 96.01$ | $2.67 \pm 1.27$ | $0.88 \pm 0.06$ | $6.75 \pm 1.12$ |

| Training Set | MAUBFC-rPPG [22] | | | | |
| Testing Set | UBFC-Phys [27] | | | | |
| Metric (± Std. Err.) | MAE↓ | RMSE ↓ | MAPE↓ | $\rho$ ↑ | SNR ↑ |
| --- | --- | --- | --- | --- | --- |
| **Supervised** | | | | | |
| TS-CAN [6] | $5.03 \pm 0.67$ | $8.39 \pm 18.26$ | $6.36 \pm 0.90$ | $0.75 \pm 0.07$ | $-1.15 \pm 0.81$ |
| PhysNet (Normalized) [5] | $5.51 \pm 0.88$ | $0.44 \pm 37.65$ | $7.50 \pm 1.32$ | $0.68 \pm 0.07$ | $-0.57 \pm 1.08$ |
| DeepPhys [35] | $4.95 \pm 0.67$ | $8.37 \pm 21.53$ | $6.26 \pm 0.90$ | $0.75 \pm 0.07$ | $-0.78 \pm 0.85$ |
| EfficientPhys-C [36] | $4.80 \pm 0.58$ | $7.52 \pm 15.02$ | $6.10 \pm 0.79$ | $0.79 \pm 0.06$ | $-0.87 \pm 0.86$ |

| Training Set | MAUBFC-rPPG [22] | | | | |
| Testing Set | MMPD [25] | | | | |
| Metric (± Std. Err.) | MAE↓ | RMSE ↓ | MAPE↓ | $\rho$ ↑ | SNR ↑ |
| --- | --- | --- | --- | --- | --- |
| **Supervised** | | | | | |
| TS-CAN [6] | $12.59 \pm 0.62$ | $20.23 \pm 31.27$ | $13.77 \pm 0.62$ | $0.23 \pm 0.04$ | $-9.19 \pm 0.29$ |
| PhysNet (Normalized) [5] | $10.68 \pm 0.49$ | $16.56 \pm 19.72$ | $14.01 \pm 0.72$ | $0.32 \pm 0.04$ | $-9.28 \pm 0.21$ |
| DeepPhys [35] | $12.71 \pm 0.65$ | $21.04 \pm 35.40$ | $13.70 \pm 0.64$ | $0.21 \pm 0.04$ | $-8.85 \pm 0.31$ |
| EfficientPhys-C [36] | $13.42 \pm 0.66$ | $21.64 \pm 35.46$ | $14.52 \pm 0.65$ | $0.14 \pm 0.04$ | $-9.20 \pm 0.31$ |

MAE = Mean Absolute Error in HR estimation (Beats/Min), RMSE = Root Mean Square Error in HR estimation (Beats/Min), MAPE = Mean Percentage Error (%), $\rho$ = Pearson Correlation in HR estimation, SNR = Signal-to-Noise Ratio (dB) when comparing predicted spectrum to ground truth spectrum.

Table 10: **Training with Pseudo Labels.** For the supervised methods we show results training with the (entire) BP4D+ [26] dataset, using POS [31] derived pseudo training labels.

| Training Set | BP4D+[26] with POS Pseudo Labels | | | | |
| Testing Set | UBFC-rPPG [22] | | | | |
| Metric (± Std. Err.) | MAE↓ | RMSE ↓ | MAPE↓ | $\rho$ ↑ | SNR ↑ |
| --- | --- | --- | --- | --- | --- |
| **Supervised** | | | | | |
| TS-CAN [6] | $4.69 \pm 1.88$ | $13.04 \pm 100.15$ | $4.51 \pm 1.65$ | $0.78 \pm 0.10$ | $0.01 \pm 1.27$ |
| PhysNet(Normalized) [5] | $1.78 \pm 0.67$ | $4.68 \pm 11.94$ | $1.92 \pm 0.72$ | $0.96 \pm 0.04$ | $1.24 \pm 1.08$ |
| DeepPhys [35] | $2.74 \pm 0.96$ | $6.78 \pm 27.43$ | $2.81 \pm 0.91$ | $0.93 \pm 0.06$ | $-0.22 \pm 1.33$ |
| EfficientPhys-C [36] | $2.43 \pm 1.29$ | $8.68 \pm 67.51$ | $2.52 \pm 1.20$ | $0.90 \pm 0.07$ | $0.39 \pm 1.27$ |

| Training Set | BP4D+[26] with POS Pseudo Labels | | | | |
| Testing Set | PURE [23] | | | | |
| Metric (± Std. Err.) | MAE↓ | RMSE ↓ | MAPE↓ | $\rho$ ↑ | SNR ↑ |
| --- | --- | --- | --- | --- | --- |
| **Supervised** | | | | | |
| TS-CAN [6] | $1.29 \pm 0.76$ | $6.00 \pm 33.74$ | $1.60 \pm 0.86$ | $0.97 \pm 0.03$ | $8.61 \pm 1.02$ |
| PhysNet(Normalized) [5] | $3.69 \pm 1.46$ | $11.79 \pm 64.42$ | $7.35 \pm 3.01$ | $0.88 \pm 0.06$ | $8.33 \pm 0.06$ |
| DeepPhys [35] | $2.47 \pm 1.41$ | $11.11 \pm 93.02$ | $2.49 \pm 1.21$ | $0.89 \pm 0.061$ | $7.32 \pm 1.09$ |
| EfficientPhys-C [36] | $3.59 \pm 1.84$ | $14.55 \pm 135.51$ | $3.27 \pm 1.50$ | $0.80 \pm 0.08$ | $7.48 \pm 1.15$ |

MAE = Mean Absolute Error in HR estimation (Beats/Min), RMSE = Root Mean Square Error in HR estimation (Beats/Min), MAPE = Mean Percentage Error (%), $\rho$ = Pearson Correlation in HR estimation, SNR = Signal-to-Noise Ratio (dB) when comparing predicted spectrum to ground truth spectrum.

Table 11: **Full 3-Fold Multitasking Results.** For the BigSmall [41] method we show the full 3-fold results for multi-tasking PPG, respiration, and action unit classification; training and evaluating on the BP4D+ [26] (AU subset) dataset, using POS [31] derived pseudo training PPG labels.

| | Training Set
Testing Set
Fold | BP4D+[26]
BP4D+[26]
Fold 1 | | Fold 2 | | Fold 3 | |
|---|---|---|---|---|---|---|---|
| **Heart** | MAE↓ | $4.24 \pm 0.73$ | | $2.91 \pm 0.49$ | | $2.54 \pm 0.48$ | |
| **(Metric ± Std. Err.↓)** | RMSE↓ | $10.76 \pm 33.20$ | | $7.26 \pm 13.90$ | | $7.06 \pm 16.67$ | |
| | MAPE↓ | $4.55 \pm 0.74$ | | $3.22 \pm 0.53$ | | $2.75 \pm 0.49$ | |
| | $\rho$↑ | $0.68 \pm 0.05$ | | $0.90 \pm 0.03$ | | $0.91 \pm 0.03$ | |
| | SNR↑ | $3.85 \pm 0.69$ | | $6.27 \pm 0.67$ | | $6.53 \pm 0.63$ | |
| **Respiration** | MAE↓ | $5.28 \pm 0.31$ | | $4.96 \pm 0.33$ | | $5.34 \pm 0.35$ | |
| **(Metric ± Std. Err.↓)** | RMSE↓ | $6.74 \pm 4.38$ | | $6.67 \pm 4.96$ | | $7.18 \pm 5.12$ | |
| | MAPE↓ | $24.41 \pm 1.55$ | | $25.30 \pm 2.08$ | | $29.14 \pm 2.72$ | |
| | $\rho$↑ | $0.15 \pm 0.07$ | | $0.16 \pm 0.72$ | | $0.12 \pm 0.07$ | |
| | SNR↑ | $7.69 \pm 0.64$ | | $10.53 \pm 0.75$ | | $9.34 \pm 0.64$ | |
| **Facial Action (AU)** | AU01 | 18.62 | 11.04 | 18.88 | 11.34 | 24.32 | 16.43 |
| **(F1↑, Prec.↑)** | AU02 | 20.76 | 12.73 | 18.28 | 10.89 | 15.46 | 9.07 |
| | AU04 | 12.57 | 8.08 | 11.48 | 7.85 | 14.43 | 8.63 |
| | AU06 | 66.73 | 66.58 | 64.71 | 61.09 | 76.44 | 79.20 |
| | AU07 | 74.86 | 78.68 | 70.08 | 75.10 | 75.58 | 86.34 |
| | AU10 | 74.92 | 77.32 | 70.09 | 74.48 | 82.09 | 90.34 |
| | AU12 | 72.69 | 70.79 | 67.75 | 68.54 | 80.96 | 88.02 |
| | AU14 | 67.21 | 72.84 | 70.18 | 69.11 | 66.73 | 70.93 |
| | AU15 | 22.56 | 13.91 | 22.33 | 13.38 | 29.64 | 22.13 |
| | AU17 | 25.77 | 18.01 | 20.95 | 12.45 | 38.17 | 28.06 |
| | AU23 | 34.64 | 27.41 | 34.21 | 24.19 | 40.68 | 28.76 |
| | AU24 | 7.00 | 3.71 | 10.70 | 6.20 | 19.03 | 10.81 |
| **Facial Action (AU)** | F1↑ | 41.53 | | 39.97 | | 46.96 | |
| **(Metric Mean)** | Prec.↑ | 36.42 | | 36.22 | | 44.89 | |
| | Acc. (%)↑ | 61.91 | | 62.42 | | 72.83 | |

For HR estimation, MAE = Mean Absolute Error, RMSE = Root Mean Square Error, MAPE = Mean Percentage Error (%), $\rho$ = Pearson Correlation, SNR = Signal-to-Noise Ratio (dB) when comparing predicted spectrum to ground truth spectrum. For AU classification F1 = harmonic mean of precision and recall, Prec. = precision, Acc. = accuracy.

## H    UBFC-Phys Video Exclusion

For evaluation of the UBFC-Phys [27] dataset in our main paper and by default in our toolbox, we utilized all three tasks and the same subject exclusion, or conversely sub-selection, list provided by the authors of the dataset in the second supplementary material of their paper [27]. Based on the aforementioned supplementary material, we eliminated 14 subjects (s3, s8, s9, s26, s28, s30, s31, s32, s33, s40, s52, s53, s54, s56) for the rest task (T1), 30 subjects (s1, s4, s6, s8, s9, s11, s12, s13, s14, s19, s21, s22, s25, s26, s27, s28, s31, s32, s33, s35, s38, s39, s41, s42, s45, s47, s48, s52, s53, s55) for the speech task (T2), and 23 subjects (s5, s8, s9, s10, s13, s14, s17, s22, s25, s26, s28, s30, s32, s33, s35, s37, s40, s47, s48, s49, s50, s52, s53) for the arithmetic task (T3).

In our toolbox, video exclusion is achieved using dataset filtering criteria specified in the config file. Specifically, an exclusion list or a task selection list can be provided to respectively exclude videos from being included or to select specific tasks as a part of a dataset.

## I    Multitasking Training and Evaluation Details

To show how this toolbox may be extended for physiological multitasking, we implement BigSmall [41] a model that multitasks PPG, respiration, and facial action. Here we reiterate information from [41], with slight modifications, for clarification.

### I.1    Cross Validation Subject Folds

**Fold 1:** F003, F004, F005, F006, F009, F017, F022, F028, F029, F031, F032, F033, F038, F044, F047, F048, F052, F053, F055, F061, F063, F067, F068, F074, F075, F076, F081, M003, M005, M006, M009, M012, M019, M025, M026, M028, M031, M036, M037, M040, M046, M047, M049, M051, M054, M056.

**Fold 2:** F001, F002, F008, F018, F021, F025, F026, F035, F036, F037, F039, F040, F041, F042, F046, F049, F057, F058, F060, F062, F064, F066, F070, F071, F072, F073, F077, F082, M001, M002, M007, M013, M014, M016, M022, M023, M024, M027, M029, M030, M034, M035, M041, M042, M043, M048, M055.

**Fold 3:** F007, F010, F011, F012, F013, F014, F015, F016, F019, F020, F023, F024, F027, F030, F034, F043, F045, F050, F051, F054, F056, F059, F065, F069, F078, F079, F080, M004, M008, M010, M011, M015, M017, M018, M020, M021, M032, M033, M038, M039, M044, M045, M050, M052, M053, M057, M058.

### I.2    AU Subset

The AU subset used for model training and evaluation (in this toolbox) is made up of dataset subset which contains action unit labels. This consists of approximately 20 seconds worth of data from the following tasks for each subject: T1, T6, T7, T8.

### I.3    Subject Fold Splits

[41] is evaluated using 3 fold cross validation, where the folds are comprised of trials from mutually exclusive subjects in the dataset. These subject-wise folds are outlined below.

## J    Additional Features
### J.1    Pre-processed Data Visualization

Pre-processing is an important aspect of the rPPG task that we hope to help standardize using our toolbox. It is advantageous to be able to quickly visualize and visually evaluate pre-processed image data and ground truth signals. Image data in particular can be especially useful to observe in order to inspect the effectiveness of out-of-the-box face detection and cropping techniques used in our toolbox, and to ultimately get an idea as to how much of the face region is visible in a given video. We provide simple Jupyter Notebooks for quickly visualizing image data and ground truth signals pre-processed by our toolbox. Further details regarding these notebooks can be found in our GitHub repo and the associated README.

### J.2 Training Loss, Validation Loss, and Learning Rate Visualization

The rPPG-Toolbox assumes certain defaults across most config files for supervised methods, including a default learning rate of 0.009 used alongside the Adam [38] or AdamW [39] optimizers, a criterion such as mean squared error (MSE) loss or Negative Pearson Correlation Loss, and the 1cycle learning rate scheduler [40] are utilized for training. An exception is with BigSmall [41], which uses a default learning rate of 0.001 that remains constant throughout training. It can be valuable to visualize losses such as those involved in training or validation phases. Furthermore, it may be useful to simultaneously visualize the learning rate, especially when users stray from the defaults in order to target an optimal set of training, validation, and testing parameters for their research efforts. The toolbox's configs contain parameters that enable the visualization of the training loss, validation loss, and the learning rate for any given supervised method.

### J.3 Bland-Altman Plots

We provide Bland-Altman plots as an additional metric in the rPPG-Toolbox. Users can enable the plots via an evaluation parameter in the config file, and will be given further options to configure the plots as the toolbox is refined and expanded. For more details, please refer to the GitHub repo and the associated README.

### J.4 Motion Analysis

We also provide scripts that leverage OpenFace [44] for extracting, visualizing, and analyzing motion in rPPG video datasets. Specifically, we include a Python script to convert datasets into the .mp4 format for subsequent analysis by OpenFace, a shell script that leverages OpenFace to perform both rigid and non-rigid head motion analysis, and a separate Python script that plots exemplar plots that showcase comparisons of motion between different datasets. Further details can be found in our GitHub repo and the associated README.

