# OpenReview forum: "rPPG-Toolbox: Deep Remote PPG Toolbox"
_NeurIPS.cc/2023/Track/Datasets_and_Benchmarks — NeurIPS 2023 Datasets and Benchmarks Poster_

### Official Review · Reviewer_fJMT · 2023-06-29
**A toolbox and benchmark tests for multiple models**

**Rating:** 8
**Confidence:** 4
**Clarity:** Yes, the expression of this article i…

**Strengths:**

1. Compared to previous work, the proposed rPPG-Toolbox supports neural network training and testing, which may be the first benchmarking tool to support neural network training.

2. Reproduced several unsupervised methods, confirmed early work, and provided practical examples.

3. Trained and tested multiple neural networks, including some of the latest multimodal works.

4. Provided a self-supervised training method using pseudo-labels.

5. The toolbox unifies preprocessing and post-processing.

6. The toolbox has a large-scale community and attention.

**Additional Feedback:**

I highly appreciate the authors' contributions to the community. In the past, due to the need for a series of preprocessing and post-processing steps in rPPG algorithms, which were complex and numerous, many algorithms were difficult to reproduce. The authors have reproduced a series of models and conducted detailed benchmark tests, which is an important contribution.

For the community, fairness in benchmark testing is equally important. The authors have reproduced 5 models (4 rPPG models and 1 multimodal model), all with similar structures except for PhysNet and from the same team. Reproducing other types of models from other teams is as important as reproducing one's own model.

For this paper, my main concern is that it does not demonstrate the reproducibility of end-to-end models. Although the authors claim to use a model with DiffNormalized inputs as an end-to-end model, the community recognizes more of those using Raw inputs as end-to-end models. Even though EfficientPhys did not use DiffNormalized input, in reality this process was just moved inside the model and completed with Diff+BatchNorm, which is essentially similar.

I sincerely hope that the authors can reproduce PhysNet and PhysFormer according to the original text. Although the training and testing codes of these two models are not fully open source, as long as the corresponding preprocessing and postprocessing processes are correctly configured, it is entirely feasible to reproduce them.

**Correctness:**

This paper may have issues with correctness, mainly due to the inconsistency between the reproduction of PhysNet and the description in [2]. Although it is claimed in the text that [2] does not specify an input format, the original text is not as claimed by the authors. In Section 3.1 of [2], "The overall architecture of PhysNet is shown in Figure 2. The input of the network is T-frame face images with RGB channels.", The author clearly states that model inputs are RGB images, but this paper's reproduction does not use RGB images, so comparisons between PhysNet and other baselines may be unfair.

In Section 3.3, line 211, this toolbox provides three data types: DiffNormalized, Standardized and Raw. However, none of these data types are suitable for true end-to-end models (i.e., models that do not require difference inputs), which use Raw inputs and Normalized outputs instead. This approach was taken because different pulse meters may have different waveform data ranges while different cameras have identical RGB ranges.

For specific implementation details refer to https://github.com/ZitongYu/PhysNet, It should be noted that the normalize here only applies to the case where batch_size=1. To use it in rPPG-ToolBox, modifications should be made to ensure that each label is normalized separately.

According to the source code of rPPG-Toolbox, lines 66 and 67 of https://github.com/ubicomplab/rPPG-Toolbox/blob/main/neural_methods/trainer/PhysnetTrainer.py, the authors noticed this during the training phase (but still did not handle it correctly), and did not perform Normalize in the testing phase, which would result in poor waveform concatenation effects.

Waveform concatenation is a crucial step for models using Normalized output (or Raw output). Unlike models using DiffNorm output, the waveform-direct model outputs usually cannot be concatenated directly, especially for models using Neg_pearson loss. This is because it does not impose any constraints on the waveform trend, which may lead to deviations at the beginning and end of each segment of output, resulting in periodic noise in the concatenated signal. Since all models in this paper use DiffNorm outputs and concatenation problems are unique to PhysNet-like models that directly output waveforms, it cannot be determined whether the authors have correctly handled this issue.

There are some issues with the rPPG-Toolbox in terms of dataset processing.

PhysNet should use Raw input, however, Raw input generally refers to RGB images stored as fp numbers between 0\~1. In the datasets supported by rPPG-Toolbox, SCAMPS and MMPD directly store videos as fp numbers between 0\~1. However, other datasets do not. The authors should have scaled all values to 0\~255 which will not affect models using DiffNormalized inputs; after normalization this difference is offset. But Raw input is not like this; it must have a unified numerical range and it's usually 0\~1 rather than 0\~255.

[2] Yu, Z., Li, X., & Zhao, G. (2019). Remote photoplethysmograph signal measurement from facial videos using spatio-temporal networks. arXiv preprint arXiv:1905.02419.

**Documentation:**

This work is open source, so I believe all the details are sufficient and the results can be reproduced.

**Ethics:**

This work does not involve new datasets, so I have no ethical concerns. The authors chose the Responsible AI License to limit potential AI ethical risks, which further ensures this point.

**Limitations:**

The paper introduces the possible limitations, which is good. I also need to point out that face detection algorithms are also factors limiting algorithm applications. This toolbox uses a simple Haar cascade detector, and the detection boxes output by this type of detector will shake violently in the video. As a result, the toolbox has to use fixed detection boxes in some cases; specifically, it only detects the first frame and then enlarges it by 1.5 times and fixes it. This approach is easy to implement, which is an advantage; however, there are many better modern detectors available that can be used and can try to suppress box jitter.

Some algorithms may be sensitive to detection algorithms and box jittering, limiting their performance; moreover, practices in code will also limit experiments on videos with significant motion.

The main contribution of this paper lies in replicating neural models; however,it should be noted that all models replicated by authors use difference-normalized input instead of raw input end-to-end models (e.g., PhysFormer[1], PhysNet[2]), so there may be limitations on model types.

The paper calculated the standard error of metrics, but this is not the random seed error generally understood by the community. The former describes the credibility of metrics, while the latter describes the stability of models under different initializations. Community is more concerned about the latter, that is, whether similar results will be obtained if different random seeds are used for convergence.

**Opportunities For Improvement:**

From the perspective of the community, it is hoped that the paper can be improved in the following aspects:

1. Provide a visualization tool for BVP signals. Usually, in a video, different segments have different waveforms, and some parts may have large errors caused by sudden head movements or changes in lighting. However, analyzing this impact is not easy and requires manual waveform drawing and alignment with videos. This toolbox only provides indicators for the entire video and plots their distribution but lacks visualization of model outputs.

2. Provide HRV-related tools as HRV is more important than HR in many cases, especially in affective computing. This toolbox has fully evaluated HR indicators but lacks evaluation of HRV.

3. On Github's readme page, this toolbox provides a way to add new datasets but does not provide a way to add new algorithms which are essential since currently supported neural algorithms are limited.

4. It is suggested to reproduce the latest models, such as PhysFormer[1], which is a recent work on CVPR. The code is open source, and the author should consider reproducing this model.

5. In benchmark test results presentation (Table 2), more representative metrics should be chosen like MAPE+RMSE because MAE calculation method is very similar to MAPE; simultaneously displaying both metrics does not thoroughly evaluate models. Replacing one with RMSE better reflects extreme error situations; it's easy since related data exists within supporting materials.

6. The article uses Butterworth filter for band-pass filtering algorithm output signals between 0.75\~2.5 Hz corresponding heart rate range 45\~150 BPM which might be too small. In real-life scenarios heart rates could exceed 150 (e.g., excitement or exercise) or fall below 45 BPM, PURE dataset reportedly contains samples below 40 BPM. Such narrow band-pass filters usually suppress occurrence of significant errors while also limiting assessment applicability scenarios. community prefers fairer & comprehensive post-processing procedures e.g., using 0.5\~4 Hz band-pass filters.

7. It is suggested to add a summary of the experimental data, such as how to choose the rPPG model in different scenarios based on the experimental data, or how to select an appropriate dataset for training the model.

[1] Yu, Z., Shen, Y., Shi, J., Zhao, H., Torr, P. H., & Zhao, G. (2022). PhysFormer: facial video-based physiological measurement with temporal difference transformer. In Proceedings of the IEEE/CVF Conference on Computer Vision and Pattern Recognition (pp. 4186-4196).

**Relation To Prior Work:**

This article summarizes many previous algorithms and has obvious advantages compared to the previous rPPG framework, supporting the training and testing of neural networks.

**Summary And Contributions:**

This paper introduces rPPG-Toolbox, an integrated tool for training and testing that supports multiple models and datasets, and provides experimental results of various models on different datasets. The main contribution is to provide a more convenient research tool for the community, which can save time in reproducing models. In addition, benchmark tests have been conducted on previous models to verify their performance. It is an excellent research tool.

---

> ### Author Response · Authors · 2023-08-20
>
> Thanks very much for your constructive feedback and valuable comments on our PhysNet implementation. Here are the responses for each of your questions and suggestions.
>
> **Question**: model output visualization.
>
> **Response**: In fact, our toolbox does support visualization of model inputs and now we support visualize model outputs as well. We apologize that we did not point out in the paper due to the page limit. We added it in the README file in our Github repo. Please refer to the section on Visualization of Preprocessed Data in our repo’s README.
>
> **Question**:  HRV feature in the toolbox
>
> **Response**: It is a great idea to add HRV support to our toolbox. We will certainly add that to our roadmap and list it in our limitation section. We included this as one of our future directions in Supplementary Materials - Section E.
>
>
> **Question**: It is suggested to reproduce the latest models, such as PhysFormer[1], which is a recent work on CVPR. The code is open source, and the author should consider reproducing this model.
>
> **Response**: We have added PhysFormer to our supported methods in this rebuttal. Please see the new results in Table 2.
>
>
> **Question**: Why MAPE and MAE?
>
> **Response**: In fact MAE and MAPE are two different metrics in the space. Currently, MAE is the most popular metric that the community is referring to as it represents the absolute HR error. However, MAPE is the metric that the FDA and commercial applications pay attention to. More specifically, MAPE of $<$ 5\% is the recommended error threshold from American National Standards Institute (ANSI) and Consumer Technology Association (CTA) [https://shop.cta.tech/products/physical-activity-monitoring-for-heart-rate]. Thus we prioritized these two metrics in the paper but we do value the importance of other metrics. Other metrics (including, MAPE, RMSE, MAPE, Pearson and SNR) are reported in our supplementary materials.
>
> **Question**: band-pass parameters in PURE.
>
> **Response**: This is a good point. One of contributions in this toolbox is the standardization for the rPPG community. Fine-tuning these filtering parameters can potentially also improve the performance in other datasets as well. Thus, we believe using one set of filtering parameters for all the models and datasets is a fairer approach. However, users can easily adjust these parameters in the toolbox.
>
> **Question**: network recommendation.
>
> Response: This is a good point. We added a recommendation section in the supplementary materials [Section G - Network Recommendation].
>
> **Question**: face detection
>
> **Response**: This is a good point, however we argue that conducting face detection on every frame in the motion videos could introduce additional noise. Thus, we believe cropping the first face and leaving the rest of the work to neural networks to handle is a more promising direction.
>
> **Continue our response in the next thread**

---

> > ### Author Response · Authors · 2023-08-20
> > **Continued responses**
> >
> > **Question**: The main contribution of this paper lies in replicating neural models; however,it should be noted that all models replicated by authors use difference-normalized input instead of raw input end-to-end models (e.g., PhysFormer[1], PhysNet[2]), so there may be limitations on model types.
> >
> > **Response**: Based on our literature survey, raw and differenced normalized frames are two main inputs to the rPPG models currently. However, we acknowledge there are more variant models and we envision adding these new architectures to the toolbox with the research community together.  We added a section regarding our future plans in  [Supplementary Materials - Section E].
> >
> > **Question**: issues with random seed.
> >
> > **Response**: our toolbox already supported setting a fixed random seed across all the models and experiments, and it should help the community to reproduce and standardize the experiments.
> >
> >
> > **Question**: Implementation of PhysNet.
> >
> > **Response**: Thanks for your kind suggestions. We updated the PhysNet network implementation on this PR [https://github.com/ubicomplab/rPPG-Toolbox/pull/197/files]. The input format is raw frames now. The results are updated in Table 2 in the main paper. More concreted, we did two things: 1) changing the raw input range from 0-255 to 0-1; 2) adding normalization to the model output. However, with these changes, PhysNet was not able to achieve results that are close to SOTA in UBFC and PURE.  We further fine-tuned the training parameters (changed learning rate from 9e-3 to 9e-2) and post-processing bandpass filter parameters (set bandpass filtering parameters to [0.5,2.5], detrend value to 200). However, we don’t recommend using these network specified parameters as we aimed to provide a standardized training infrastructure for all the neural networks.
> >
> > All in all, we concluded that the most important piece that is missed in the original PhysNet implementation is the normalization of model output. We further tested the updated architecture with both Raw and DiffNorm inputs as shown in [Supplementary Materials - Section G Investigation of PhysNet]. We are still working on generating final numbers for UBFC-Phys and MMPD. We will add these numbers in the next few days and keep you posted.
> >
> > We appreciate your great advice and suggestions again.
> >
> >
> > **Question**: adding more model architecture and end-to-end models.
> >
> > **Response**: We added one architecture (PhysFormer) supported in this toolbox during the rebuttal period. This toolbox aims to provide an open-source toolbox to the research community to standardize and benchmark rPPG methods and datasets such that more researchers can contribute to this toolbox. As we mentioned in the future plan section in supplementary materials, we will have a plan to encourage the community to engage into this toolbox moving forward.  Regarding your question about the “end-to-end”, that was claimed in EfficientPhys and it is not the focus of this toolbox paper. The “end-to-end” claim in this paper is for we propose a toolbox supporting data processing, loading, model training, evaluation, visualization and analysis for the rPPG research community.

---

> > > ### Author Response · Authors · 2023-08-20
> > > **Updates on PhysNet**
> > >
> > > Dear Reviewer,
> > >
> > > We have updated the paper with latest PhysNet and PhysFormer results. All modifications are highlighted as blue text in the main paper and supplementary materials. Please let us know if you have additional comments or questions. We would appreciate if you can update your rating accordingly if your questions are being addressed.

---

> > > > ### Author Response · Authors · 2023-08-24
> > > > **Rebuttal**
> > > >
> > > > Dear Reviewer fJMT,
> > > >
> > > > I hope this message finds you well. Firstly, I would like to express my gratitude for the time and effort you dedicated to reviewing our paper. Your feedback has been invaluable in helping improve the quality of my work.
> > > >
> > > > I recently submitted a rebuttal addressing the concerns and suggestions you raised. I would greatly appreciate it if you could take a moment to review my responses and share your thoughts and update your rating. Understanding your perspective further will be beneficial for the finalization of the paper. I understand that you have many commitments and I truly appreciate the time you've already devoted.
> > > >
> > > > Thank you again for your valuable insights and time. I look forward to hearing from you soon.

---

> > > > > ### Author Response · Authors · 2023-08-28
> > > > > **Reminder**
> > > > >
> > > > > Dear Reviewer,
> > > > >
> > > > > This is another gentle reminder to see if you have any other additional feedback as we are approaching the last day of the discussion period (Aug 29). We would really appreciate if you can read our response and update your review accordingly. Thanks again for your time.

---

> > > ### Comment · Reviewer_fJMT · 2023-08-28
> > > **My main concerns have been addressed.**
> > >
> > > I adjusted the score from 6 to 8 to recommend acceptance of this paper.
> > >
> > > My main concerns have been addressed.  However, the authors still have some work to do, especially what they mentioned "we aimed to provide a standardized training infrastructure for all the neural networks." PhysNet needs additional hyperparameter fine-tuning mainly because the authors did not specify an output standard, that is, the BVP signals output by the model did not use a unified normalization method. This leads to different standard deviations of BVP signals output by different models, and some post-processing methods are sensitive to these parameters. The authors still need to make improvements in structure; otherwise it may be necessary to apply different parameters for different models in order achieve their best performance which can be cumbersome. But I no longer doubt about correctness of this work.

---

### Official Review · Reviewer_Up8R · 2023-07-21
**The rPPG-Toolbox is comprehensive and valuable, with room for broader impact and ethical considerations.**

**Rating:** 8
**Confidence:** 3

**Strengths:**

Significance of the contribution:
The rPPG-Toolbox presents a comprehensive solution for rPPG research, covering pre-processing, supervised and unsupervised machine learning, and post-processing stages. Its end-to-end capabilities empower researchers to seamlessly build upon prior work and ensure fair evaluations and comparisons of different methods.

Relevance to the broader research community
The toolbox's capability to support diverse rPPG research requirements is highly relevant to the broader research community. It empowers researchers in the field to explore and apply various approaches that align with their specific application needs.

Quality of the research
The rPPG-Toolbox demonstrates commendable quality, with comprehensive toolbox description.

Ethical and social implications
The paper discusses the potential impact of pervasive measurement on privacy and security, emphasizing the need for responsible technology use. To ensure responsible usage, the authors have released the rPPG-Toolbox with a Responsible AI License, restricting negative and unintended applications. Besides that, the paper also highlights highlights the benefits of the technology, such as enhancing accessibility and comfort for essential cardiac measurements and utilizing everyday devices with cameras as scalable health sensors.


**Additional Feedback:**

I appreciate the efforts made by the authors in creating the useful rPPG-Toolbox. The authors have done an excellent job in providing a well-described toolbox for rPPG research.

A few suggestions:
- it would be beneficial to include more discussions on practical demonstrations and real-world deployment scenarios to make the toolbox more user-friendly for researchers in real-world settings.
- It would be valuable to see the authors explore and address potential ethical implications beyond the responsible AI license, particularly regarding privacy concerns when using camera-based health monitoring technologies.

Some questions:
- Can you elaborate on any future plans to address the limitations mentioned in the paper, particularly in supporting pre-training techniques and practical deployment scenarios for the rPPG-Toolbox?
- How do you plan to ensure continuous updates and maintenance of the rPPG-Toolbox to keep it up-to-date and relevant for the research community?

**Clarity:**

The paper is well written and well structured, with clear headings and subheadings that make it easy to navigate.

**Correctness:**

The evaluation methods and experimeent design presented in the paper is appropriate and well-performed. The authors clearly described the rPPG-Toolbox and its features, followed by a comprehensive evaluation with multiple public datasets.

**Documentation:**

The paper demonstrates a robust benchmark design with sufficient detail to support reproducibility. The main experimental results, including code, data, and instructions, are readily available in the supplemental material or accessible through their Github link.

**Ethics:**

No.

**Limitations:**

The authors have made efforts to address limitations and potential negative societal impacts of their work, but further improvement is possible. An additional suggestion is to discuss potential applications of remote photoplethysmography beyond healthcare, such as in sports or entertainment, to expand its impact on the broader research community.






**Opportunities For Improvement:**

Significance of the contribution:
The rPPG-Toolbox lacks support for common pre-training techniques like contrastive learning, limiting its use for interested researchers. An opportunity for improvement is to expand the toolbox's capabilities, enabling various pre-training techniques and widening its applicability to diverse research needs.

Relevance to the broader research community:
The rPPG-Toolbox holds value for remote photoplethysmography researchers, but its relevance to other domains may be limited. An opportunity for improvement lies in exploring ways to adapt and extend the toolbox's functionalities, making it applicable to diverse research areas and broadening its appeal to a wider research community.

Quality of the research:
The research is of high quality, an opportunity for improvement lies in discussing the practical demonstration and deployment for real-world applications. By considering practical deployment scenarios and focusing on user-friendly implementations, the toolbox's usability for researchers in real-world settings can be further enhanced.

Ethical and social implications:
The research addresses ethical and social implications to some extent through a responsible AI license.The paper could be further improved by addressing privacy concerns if the camera captures sensitive information beyond the intended health measurement.

**Relation To Prior Work:**

The paper presents a thorough comparison of the rPPG-Toolbox with existing rPPG toolboxes such as iPhys-Toolbox and PPG-I Toolbox. The authors highlight the unique features of the rPPG-Toolbox, which include support for both supervised and unsupervised rPPG models, compatibility with public benchmark datasets, and data augmentation and systematic evaluation tools. Being an end-to-end solution, it encompasses pre-processing, training, and evaluation tools, setting it apart from other toolboxes.

**Summary And Contributions:**

The rPPG-Toolbox is an end-to-end toolbox with code for pre-processing multiple public datasets, supervised and unsupervised machine learning methods, and post-processing and evaluation tools.The toolbox aims to facilitate effective research building upon existing work and enable fair evaluation and comparison of methods on remote photoplethysmography (rPPG).

The contributions include supporting multiple public datasets and implementing a diverse range of unsupervised and supervised machine learning methods.

---

> ### Author Response · Authors · 2023-08-20
> **Responses**
>
> Thank you for your valuable feedback. Here are our responses and modifications based on your feedback.
>
> **Question**: Significance of the contribution: The rPPG-Toolbox lacks support for common pre-training techniques like contrastive learning, limiting its use for interested researchers. An opportunity for improvement is to expand the toolbox's capabilities, enabling various pre-training techniques and widening its applicability to diverse research needs.
>
> **Response**: The focus on this toolbox is currently on supervised neural methods and unsupervised signal processing methods. Supporting unsupervised training is in our roadmap and is already listed as one of our limitations. We added an extended future direction section [Supplementary Materials - Section E]
>
> **Question**: Relevance to the broader research community: The rPPG-Toolbox holds value for remote photoplethysmography researchers, but its relevance to other domains may be limited. An opportunity for improvement lies in exploring ways to adapt and extend the toolbox's functionalities, making it applicable to diverse research areas and broadening its appeal to a wider research community.
>
> **Response**: Thanks for your feedback. This is a good suggestion and that is why we also included BigSmall [41] in the toolbox and extended the toolbox to support both PPG, respiration signal, facial action units. We believe more vital signs will be extracted from videos and will be happy to support them in the later updates. We included this as one of our future directions in [Supplementary Materials - Section E]
>
> **Question**: Quality of the research: The research is of high quality, an opportunity for improvement lies in discussing the practical demonstration and deployment for real-world applications. By considering practical deployment scenarios and focusing on user-friendly implementations, the toolbox's usability for researchers in real-world settings can be further enhanced.
>
> **Response**: We added a new section to discuss potential applications of rPPG in [Supplementary Materials - Section C].
>
> **Question**: Ethical and social implications: The research addresses ethical and social implications to some extent through a responsible AI license.The paper could be further improved by addressing privacy concerns if the camera captures sensitive information beyond the intended health measurement.
>
> **Response**: We value privacy in the field of camera health sensing, and that is the reason we supported TS-CAN and EfficientPhys which were both designed with on-device processing in mind. There are several reasons that on-device processing is desirable, user privacy is one of them. Due to the page limit, we added our privacy concerns in Supplementary Materials - Section D.
>
>
> **More minor questions**:
>
> **Question**: Can you elaborate on any future plans to address the limitations mentioned in the paper, particularly in supporting pre-training techniques and practical deployment scenarios for the rPPG-Toolbox? How do you plan to ensure continuous updates and maintenance of the rPPG-Toolbox to keep it up-to-date and relevant for the research community?
>
>
> **Response**: Supporting and adding unsupervised training is our highest priority. We plan to advertise this toolbox and recruit more open-source contributors to this project to continue updating it with new datasets and algorithms. Moreover, we plan to hold a public rPPG competition based on this toolbox and attract more researchers to release their algorithms and datasets in this toolbox. We added a section regarding our future plans in Supplementary Materials - Section E.
>
> We would appreciate it if you can update your rating based on our modifications.

---

> > ### Author Response · Authors · 2023-08-24
> >
> > Dear Reviewer Up8R,
> >
> > I hope this message finds you well. Firstly, I would like to express my gratitude for the time and effort you dedicated to reviewing our paper. Your feedback has been invaluable in helping improve the quality of my work.
> >
> > I recently submitted a rebuttal addressing the concerns and suggestions you raised. I would greatly appreciate it if you could take a moment to review my responses and share your thoughts and update your rating. Understanding your perspective further will be beneficial for the finalization of the paper. I understand that you have many commitments and I truly appreciate the time you've already devoted.
> >
> > Thank you again for your valuable insights and time. I look forward to hearing from you soon.

---

> > > ### Author Response · Authors · 2023-08-28
> > > **Reminder**
> > >
> > > Dear Reviewer,
> > >
> > > This is another gentle reminder to see if you have any other additional feedback as we are approaching the last day of the discussion period (Aug 29). We would really appreciate if you can read our response and update your review accordingly. Thanks again for your time.

---

### Official Review · Reviewer_fuy4 · 2023-07-21
**rPPG-Toolbox: Deep Remote PPG Toolbox**

**Rating:** 6
**Confidence:** 3
**Correctness:** No new datasets or evaluation techniq…
**Clarity:** The paper is generally well written a…

**Strengths:**

- The authors present a comprehensive toolbox for the task of rPPG, composed of 6 unsupervised/4 supervised model implementations, pre-processing, and post-processing.
- The authors provide quantitative evaluations of the implemented models on several benchmark datasets.

**Additional Feedback:**

None.

**Documentation:**

The proposed rPPG toolbox and its details are well organized in the paper, including pre-processing, model implementation, and post-processing. They also provide GitHub repository about it, providing easy utilization to the users.

**Ethics:**

No.

**Limitations:**

The authors describe the limitations of their work and possible social impacts in the paper.

**Opportunities For Improvement:**

- This paper doesn't provide a new dataset nor new types of evaluation methods, feeling questionable whether it provides a great contribution in context of dataset and benchmark.
- The authors implemented a tool-box with widely-used augmentations, evaluation metrics, which is hard to claim developing a new benchmark.
- Implementation of the previous are mostly biased to the pre-developed methods of the authors.
- Also, the implemented models don’t include up-to-date methods except for [36].
- It’s not clear about how the train-test samples (L.220—224) are splitted under each dataset, does the toolbox includes the implementation of subject-independent cross validation and its analysis, which is a significant metrics in rPPG task?
- Why PhysNet [6] shows extremely poor performance particularly under PURE dataset compared to others?

**Relation To Prior Work:**

Yes.

**Summary And Contributions:**

For fair evaluation and comparison of rPPG methods, this paper develops a comprehensive toolbox, consisting of pre-processing, implementations of supervised- & non-supervised-models, and post-processing under several public datasets.

---

> ### Author Response · Authors · 2023-08-20
> **Responses**
>
> Thank you for your constructive comments. Here are our responses for your questions.
>
> **Question**: This paper doesn't provide a new dataset nor new types of evaluation methods, feeling questionable whether it provides a great contribution in context of dataset and benchmark. The authors implemented a tool-box with widely-used augmentations, evaluation metrics, which is hard to claim developing a new benchmark.
>
> **Response**: Based on the NeurIPS’s official guidance in the Datasets and Benchmarks Track, “Benchmarks on new or existing datasets, as well as benchmarking tools.” is listed [https://nips.cc/Conferences/2023/CallForDatasetsBenchmarks]. Given the official guidance values both benchmarking and benchmarking tools, we believe our paper is a good fit for the track (and a new dataset is not a requirement). First, this toolbox is the first rPPG toolbox that provides end-to-end neural network training features for rPPG research. Second, our proposed rPPG-Toolbox comprehensively benchmarks existing rPPG datasets with both modern neural and traditional unsupervised methods. We are not aware of any previous literature that includes such a comprehensive, open and transparent cross-dataset evaluation with testing on four popular datasets. Moreover, the robustness of the evaluation pipeline we provided is also unparalleled. We support five metrics (MAE, MAPE, RMSE, Pearson Coefficient, and SNR) as well as the standard error for each measurement. We also provided tools to visualize the model outputs and results. Very few papers in the rPPG research area release complete code. Therefore this toolbox introduces a framework for end-to-end training, evaluation, benchmarking and we believe it will be a significant resource for researchers hoping to systematically compare against the state-of-the-art.
>
>
>
>
> **Question**: Implementation of the previous are mostly biased to the pre-developed methods of the authors. Also, the implemented models don’t include up-to-date methods except for [36].
>
> **Response**: We extended the toolbox to support the PhysFormer [1] as part of our rebuttal. Please see the new results in Table 2 and supplementary materials.  Furthermore, we expect that the community will contribute more novel and original methods to this open toolbox in the near future.
>
> [1] Yu, Zitong, et al. "Physformer: Facial video-based physiological measurement with temporal difference transformer." Proceedings of the IEEE/CVF conference on computer vision and pattern recognition. 2022.
>
>
> **Question**: It’s not clear about how the train-test samples (L.220—224) are split under each dataset, does the toolbox includes the implementation of subject-independent cross validation and its analysis, which is a significant metrics in rPPG task?
>
> **Response**: For the experiments we conducted, we used subject-independent splits. More specifically, we used the first 80% of the subjects for training and the remaining of the 20% subjects for validation (all details for the subject IDs used for training and testing are available from the toolbox). Users can easily control the list of the subjects for training/validation/testing as it is defined in the configuration files. Moreover, our toolbox already offers a filelist system to support flexible subject-independent cross validation where users can define any subjects in the filelists for train/val/test. We added a new section in the README file at our Github Repo. Please refer to Section [Using Custom Data Splits and Custom File Lists].  To emphasize, our toolbox supports both intra- and inter- dataset train and test splits.
>
> **Question**: Why PhysNet [6] shows extremely poor performance particularly under PURE dataset compared to others?
>
> **Response**: The preprocessing and post-processing code of PhysNet were not made available, open-source, by the authors. We spent a considerable amount of time reproducing this work and were not able to achieve reasonable testing performance on PURE while training on SCAMPS. We believe that that is due to the overfitting of the 3D-CNN style of network in SCAMPS which is much easier to learn compared to real-world datasets. Moreover, after discussing this issue with the community, we added a normalization module at the end of PhysNet implementation [https://github.com/ubicomplab/rPPG-Toolbox/pull/197/files] and was able to improve the results a bit but found that it still performs poorly on PURE while training on SCAMPS (See the updated Table 2 in the main paper).  It is worth noting that this trick is not mentioned in their paper or their open-source codebase. Poor performance on PURE while training on SCAMPS was also found with PhysFormer which has large model capability compared to other 2D-CNN based models too. We are still working on generating final numbers for UBFC-Phys and MMPD. We will add these numbers in the next few days and keep you posted.

---

> > ### Author Response · Authors · 2023-08-20
> > **Updates on PhysNet**
> >
> > Dear Reviewer,
> >
> > We have updated the paper with latest PhysNet and PhysFormer results. All modifications are highlighted as blue text in the main paper and supplementary materials. Please let us know if you have additional comments or questions. We would appreciate if you can update your rating accordingly if your questions are being addressed.

---

> > > ### Author Response · Authors · 2023-08-24
> > > **Rebuttal**
> > >
> > > Dear Reviewer fuy4,
> > >
> > > I hope this message finds you well. Firstly, I would like to express my gratitude for the time and effort you dedicated to reviewing our paper. Your feedback has been invaluable in helping improve the quality of my work.
> > >
> > > I recently submitted a rebuttal addressing the concerns and suggestions you raised. I would greatly appreciate it if you could take a moment to review my responses and share your thoughts and update your rating. Understanding your perspective further will be beneficial for the finalization of the paper. I understand that you have many commitments and I truly appreciate the time you've already devoted.
> > >
> > > Thank you again for your valuable insights and time. I look forward to hearing from you soon.

---

> > > > ### Author Response · Authors · 2023-08-28
> > > > **Reminder**
> > > >
> > > > Dear Reviewer,
> > > >
> > > > This is another gentle reminder to see if you have any other additional feedback as we are approaching the last day of the discussion period (Aug 29). We would really appreciate if you can read our response and update your review accordingly. Thanks again for your time.

---

> > > > > ### Comment · Reviewer_fuy4 · 2023-08-28
> > > > >
> > > > > Dear Authors,
> > > > >
> > > > > Firstly, I would like to appreciate the considerable effort for responding to the reviews.
> > > > > The concerns I've suggested were addressed well in the modified paper. Therefore, I've upgraded my ratings from 4 to 6.
> > > > >
> > > > > However, I'd like to highlight that the developed methods are still biased towards the works related with authors, so I'll recommend the authors to complement the diversity of the developed methods in the toolbox.
> > > > >
> > > > > Thanks,

---

### Official Review · Reviewer_zpiW · 2023-07-28
**review of rPPG-Toolbox: Deep Remote PPG Toolbox**

**Rating:** 5
**Confidence:** 2
**Clarity:** The paper is written relatively well …

**Strengths:**

compared with previous toolboxes, this paper added DNN Training.

The toolbox provides support for six public datasets, facilitating researchers' access to diverse and relevant data for their experiments.

The toolbox offers implementations of five neural model architectures and six unsupervised learning methods, providing a range of options for researchers to experiment with and compare.

The toolbox supports advanced techniques like weakly supervised pseudo labels, motion augmentation, and multitask learning, empowering researchers to explore innovative approaches for improved results.

**Additional Feedback:**

NA

**Correctness:**

The claims made in the submission is correct.


**Documentation:**

There is sufficient detail on data collection and organization, availability and maintenance, and ethical and responsible use

**Ethics:**

No ethical concerns.

**Limitations:**

No limitations and potential negative impacts have been discussed in the paper.

**Opportunities For Improvement:**

 This paper made a replication of results and benchmarking rPPG models, which can be considered only minor contributions given that the datasets and methods are all originally released.

The paper assumes some prior knowledge of the field of camera-based physiological sensing and the challenges it faces. Providing a brief background or overview of the existing research could make the paper more accessible to readers who might be less familiar with the topic.

**Relation To Prior Work:**

some introduction to camera-based physiological sensing would be better for readers outside this field.

**Summary And Contributions:**

This paper discusses the vision of ubiquitous computing and its potential to embed computation into everyday objects, particularly cameras, for accurate health signal measurements. The field of camera-based physiological sensing has seen advances but lacks standardization. To address this, the paper presents an end-to-end toolbox for camera-based physiological measurement, including support for datasets, neural model architectures, evaluation pipelines, and advanced neural training, aiming to provide clear and reproducible benchmarks for the research community.

---

> ### Author Response · Authors · 2023-08-20
> **Responses**
>
> Thank you for your great feedback. Here are our responses to your comments.
>
> **Question**:  The paper assumes some prior knowledge of the field of camera-based physiological sensing and the challenges it faces. Providing a brief background or overview of the existing research could make the paper more accessible to readers who might be less familiar with the topic. No limitations and potential negative impacts have been discussed in the paper.
>
> **Response**: Theory behind rPPG and more related work. Due to the page limit we have added this content in Supplementary Materials - Section B. It is worth noting that we do have a limitation section in the main paper [Section 5], but we added an extended future direction section [Supplementary Materials - Section E] in the supplementary materials. We also added an extended broad impact section to discuss potential negative impacts, privacy concerns and potential ethical concerns in [Supplementary Materials - Section D].
>
> **Question**: This paper made a replication of results and benchmarking rPPG models, which can be considered only minor contributions given that the datasets and methods are all originally released.
>
> **Response**: Based on the NeurIPS’s official guidance in the Datasets and Benchmarks Track, “Benchmarks on new or existing datasets, as well as benchmarking tools.” is listed [https://nips.cc/Conferences/2023/CallForDatasetsBenchmarks]. Given the official guidance values both benchmarking and benchmarking tools, we believe our paper is a good fit for the track (and a new dataset is not a requirement). First, this toolbox is the first rPPG toolbox that provides end-to-end neural network training features for rPPG research. Second, our proposed rPPG-Toolbox comprehensively benchmarks existing rPPG datasets with both modern neural and traditional unsupervised methods. We are not aware of any previous literature that includes such a comprehensive, open and transparent cross-dataset evaluation with testing on four popular datasets. Moreover, the robustness of the evaluation pipeline we provided is also unparalleled. We support five metrics (MAE, MAPE, RMSE, Pearson Coefficient, and SNR) as well as the standard error for each measurement. We also provided tools to visualize the model outputs and results. Very few papers in the rPPG research area release complete code. Therefore this toolbox introduces a framework for end-to-end training, evaluation, benchmarking and we believe it will be a significant resource for researchers hoping to systematically compare against the state-of-the-art.

---

> > ### Author Response · Authors · 2023-08-24
> > **Rebuttal**
> >
> > Dear Reviewer zpiW,
> >
> > I hope this message finds you well. Firstly, I would like to express my gratitude for the time and effort you dedicated to reviewing our paper. Your feedback has been invaluable in helping improve the quality of my work.
> >
> > I recently submitted a rebuttal addressing the concerns and suggestions you raised. I would greatly appreciate it if you could take a moment to review my responses and share your thoughts and update your rating. Understanding your perspective further will be beneficial for the finalization of the paper. I understand that you have many commitments and I truly appreciate the time you've already devoted.
> >
> > Thank you again for your valuable insights and time. I look forward to hearing from you soon.

---

> > > ### Author Response · Authors · 2023-08-28
> > > **Reminder**
> > >
> > > Dear Reviewer,
> > >
> > > This is another gentle reminder to see if you have any other additional feedback as we are approaching the last day of the discussion period (Aug 29). We would really appreciate if you can read our response and update your review accordingly. Thanks again for your time.

---

### Author Response · Authors · 2023-08-20
**Overall Response**

Overall Response

Thanks for all the constructive feedback from all the reviewers. We have responded to each of you individually and we want to highlight the changes we made throughout this rebuttal process.

**PhysNet**: Regenerated PhysNet’s results based on the community’s and Reviewer fJMT’s feedback. Results are updated in Table 2 in the main paper as well as Table 2 and Table 3 in Supplementary Materials. We also conducted a series of experiments across different input formats in PhysNet and explained why we used DiffNorm frames as input. The results are included in the new Supplementary Materials - Section F.

**PhysFormer**: Since several reviewers asked us to add PhysFormer to this toolbox, we implemented it and evaluated it  across all the datasets. The results are included in Table 2 in the main paper as well as Table 2 and Table 3 in Supplementary Materials.

**rPPG Background**: Due to the page limit, we didn’t add too many details regarding rPPG background. We added a new section called “Background and Existing Research in rPPG” in Supplementary Materials - Section B to discuss rPPG background, optical basis and existing algorithms.

**Potential Applications**: We added a new section Supplementary Materials - Section C to discuss potential applications of rPPG technology.

**Future Directions**: We added a new section Supplementary Materials - Section E to discuss our future plans for this toolbox.
Broader Impact: We added a new section Supplementary Materials - Section D to discuss privacy concerns, potential negative impacts and potential ethical concerns of rPPG applications.

**Network Recommendation**: We added a new section Supplementary Materials - Section G to discuss our insights and recommendations while using these rPPG networks during deployments.

Overall, during this rebuttal, we have regenerated our PhysNet’s results, supported a new architecture (PhysFormer) and added 6 new sections in supplementary materials to address various questions and concerns. Please let us know if you have additional questions. We would appreciate it if you can update your rating as your questions are being addressed.

---

### Decision · Program_Chairs · 2023-09-22

**Decision:**

Accept (Poster)

**Comment:**

The initial reviews raised several concerns that were mostly addressed after the rebuttal! The authors are encouraged to address the concerns in the final paper.